



# The sub-adiabatic model as a concept for evaluating the representation and radiative effects of low-level clouds in a high-resolution atmospheric model

Vasileios Barlakas[1], Hartwig Deneke[1], and Andreas Macke[1]

[1]Leibniz Institute for Tropospheric Research, Leipzig, Germany

**Correspondence:** barlakas@tropos.de

**Abstract.** The realistic representation of low-level clouds, including their radiative effects, in atmospheric models remains challenging. A sensitivity study is presented to establish a conceptual approach for the evaluation of low-level clouds and their radiative impact in a highly resolved atmospheric model. Considering simulations for six case days, the analysis supports that the properties of clouds more closely match the assumptions of the sub-adiabatic rather than the vertically homogeneous cloud model, suggesting its use as basis for evaluation. For the considered cases, 95.7 % of the variance in cloud optical thickness is explained by the variance in the liquid water path, while the droplet number concentration and the sub-adiabatic fraction contribute only 3.5 % and 0.14 % to the total variance, respectively. A mean sub-adiabatic fraction of 0.45 is found, which exhibits strong inter-day variability. Applying a principal component analysis and subsequent varimax rotation to the considered set of nine properties, four dominating modes of variability are identified, which explain 98 % of the total variance. The first and second components correspond to the cloud base and top height, and to liquid water path, optical thickness, and cloud geometrical extent, respectively, while the cloud droplet number concentration and the sub-adiabatic fraction are the strongest contributors to the third and fourth components. Using idealized offline radiative transfer calculations, it is confirmed that the shortwave and longwave cloud radiative effect exhibits little sensitivity to the vertical structure of clouds. This reconfirms, based on an unprecedented large set of highly resolved vertical cloud profiles, that the cloud optical thickness and the cloud top and bottom heights are the main factors dominating the shortwave and longwave radiative effect of clouds, and should be evaluated together with radiative fluxes using observations, to attribute model deficiencies in the radiative fluxes to deficiencies in the representation of clouds. Considering the different representations of cloud microphysical processes in atmospheric models, the analysis has been further extended and the deviations between the radiative impact of the single- and double-moment schemes are assessed. Contrasting the shortwave cloud radiative effect obtained from the double-moment scheme to that of a single moment scheme, differences of about $\sim 40 \, \mathrm{W \, m^{-2}}$ and significant scatter are observed. The differences are attributable to a higher cloud albedo resulting from the high values of droplet number concentration in particular in the boundary layer predicted by the double-moment scheme, which reach median values of around $\sim 600 \, \mathrm{cm^{-3}}$.





# 1 Introduction

Clouds play a crucial role in the global energy budget and climate. One important aspect is their strong influence on the shortwave (SW) and longwave (LW) radiation budget. Despite significant progress over the past decades, the relevant processes and resulting climate feedbacks of clouds have not been fully understood, and cannot be reliably represented in climate projec-
tions (IPCC, 2013). The representation of boundary layer clouds (i.e., shallow cumulus, stratiform) is particularly problematic (Turner et al., 2007), due to their high spatio-temporal variability. In addition, the coarse resolution of general circulation models ($\sim 100\,\mathrm{km}$) is not sufficient to resolve processes taking placed at sub-grid scale, nor allows to explicitly take vertical and horizontal heterogeneity into consideration.

Clouds are characterized by complicated three-dimensional (3D) shapes with highly variable macrophysical, microphysical,
and radiative properties. Full 3D radiative transfer calculations in complex cloudy atmospheres are computationally expensive and, hence, a number of simplifications are commonly adopted for calculating their radiative effect in atmospheric models. The plane-parallel (PP) approximation is often utilized, which implies that radiative transfer simulations are conducted assuming horizontally homogeneous clouds covering a fraction of the model grid (Di Giuseppe and Tompkins, 2003; Chosson et al., 2007). One particular shortcoming of this assumption is the so-called plane-parallel albedo bias, which refers to the fact
that inhomogeneous clouds reflect less solar radiation than otherwise identical homogeneous clouds (Werner et al., 2014). To account for this bias, and to consider horizontal heterogeneities in GCMs, several correction schemes have been developed over the last years, e.g., scaling the liquid water path by a constant reduction factor, renormalization techniques, among others (e.g., Cahalan et al., 1994b; Barker, 2000; Cairns et al., 2000; Barker and Räisänen, 2004; Pincus et al., 2003; Shonk and Hogan, 2008).

The optical properties of a cloudy layer are largely determined by two of their physical properties, the liquid water content ($q_{\mathrm{L}}$) and the effective radius ($r_{\mathrm{eff}}$) (Slingo, 1989; Collins et al., 2006). The latter is mostly obtained by assuming a fixed droplet number distribution (Chosson et al., 2007). Double-moment cloud microphysical schemes, which also constrain the effective radius through prognostic equations, are only recently becoming more widespread in use.

To improve the scientific understanding of clouds and their representation in models, high-quality observations from active
(i.e., lidar and cloud radar) and passive (i.e., radiometers) instruments from both ground and space are essential. Currently, such instrumentation is available, i.e., from the Cloudnet program (Illingworth et al., 2007), the A-train constellation (Stephens et al., 2002), the geostationary satellite Meteosat Second Generation (MSG) (Roebeling et al., 2006), while upcoming missions comprise the Earth Cloud Aerosol Radiation Explorer (EarthCARE) satellite mission (Illingworth et al., 2015) and Meteosat Third Generation (MTG) (Stuhlmann et al., 2005). A variety of algorithms have been developed for inferring cloud properties
from these observations (e.g., Nakajima and King, 1990; Bennartz, 2007; Roebeling et al., 2013). However, the underlying observational techniques often rely heavily on assumptions about the cloud vertical structure.

High-resolution atmospheric models at cloud-resolving scales are another promising avenue to gain insights into cloud processes and the effects of small-scale cloud variability, and to improve their representation in GCMs. They can resolve relevant processes up to a much smaller scale ($\sim 100\,\mathrm{m}$ for Large Eddy Simulations), and can, thus, serve as basis for developing more





accurate parameterizations. Enabled by the exponential growth in computer power over the past decades, they are increasingly utilized for simulations covering larger domains and longer time periods. In contrast to observations, they also offer the opportunity to assess the interplay of all relevant state variables simultaneously, while instrumental capabilities are generally limited to a small subset, sometimes affected by large measurement uncertainties (Miller et al., 2016).

It is, however, crucial to also critically evaluate the performance of high-resolution atmospheric models with observations. Like coarse-resolution models, they include various assumptions and parameterizations, and their shortcomings need to be identified and mitigated. Given the complexity of atmospheric models and the level of detail available from the output of such models, it is, however, often a daunting task to identify the physical reasons for model shortcomings. Inconsistent or even conflicting assumptions made in observation-based products add additional complications to the evaluation of models with

observations. Examples for such assumptions include a vertically homogeneous or a sub-adiabatic cloud, which is often made in satellite retrievals (Brenguier et al., 2000; Chosson et al., 2007), or the assumption of a vertically constant cloud droplet number concentration commonly used in ground-based remote sensing of clouds, which is a significant simplification of the profiles available from in situ observations or double-moment cloud microphysical schemes.

In this work, the highly resolved ICON-LEM atmospheric model (ICOsahedral Non-hydrostatic Large-Eddy Model) is em-

ployed that was recently developed within the HD(CP)$^2$ (High Definition Clouds and Precipitation for advancing Climate Prediction) project (Dipankar et al., 2015; Heinze et al., 2017). ICON-LEM provides an unprecedented horizontal resolution of 156, 312, and 625 m covering a large domain over Germany. We introduce a conceptual approach for evaluating the representation of low-level clouds in this and other high-resolution atmospheric models, with particular focus on the correct representation of their radiative effect. A sensitivity study is conducted in order to investigate the relevance of the vertical dis-

tribution of microphysical properties for their radiative effect, aiming for the identification of suitable column-effective cloud properties for the purpose of model evaluation. The suitability of the sub-adiabatic cloud model is compared to that of the vertically homogeneous cloud model, both of which are commonly used in remote sensing. In addition, differences in cloud radiative properties arising from the availability of the cloud droplet number concentration provided by the double-moment cloud microphysical scheme of Seifert and Beheng (2006) compared to a single-moment scheme are highlighted.

## 2    Data and methods

### 2.1    ICON-LEM

The ICON unified modeling framework was co-developed by the German meteorological service (DWD) and the Max Planck institute for meteorology (MPI-M) in order to support climate research and weather forecasting. Within the HD(CP)$^2$ project, ICON was further extended towards large eddy simulations with realistic topography, open boundary conditions, and a nesting

approach with grids varying from 624 m to 312 m, and 156 m resolution. This resulted in ICON-LEM (Heinze et al., 2017). Concerning turbulence parameterization, the three-dimensional Smagorinsky scheme is employed (Dipankar et al., 2015). The activation of cloud condensation nuclei (CCN) is based on the parameterization of Seifert and Beheng (2006) and modified





in order to account for the consumption of CCNs due to their activation into cloud droplets. The CCN concentration is then parameterized following the pressure profile and the vertical velocity (Hande et al., 2016).

ICON-LEM utilizes the double-moment mixed-phase bulk microphysical parameterization scheme introduced by Seifert and Beheng (2006). Following their comprehensive description, a generalized gamma distribution is utilized to describe the

mass ($x_{\mathrm{m}}$) of hydrometeors,

$$f\left(x_{\mathrm{m}}\right) = A_{\mathrm{m}} \cdot x^{\nu} \cdot \exp\left(-B_{\mathrm{m}} \cdot x_{\mathrm{m}}^{\xi}\right). \tag{1}$$

The coefficients $\nu$, $\xi$ are constants taken from Table 1 in Seifert and Beheng (2006) while the coefficients $A_{\mathrm{m}}$ and $B_{\mathrm{m}}$ are prognostic quantities expressed by the number and mass densities (see Appendix A).

The model yields output on each of the aforementioned grids with the data stored as one-dimensional (1D) profiles, two-

(2D), and 3D snapshots (Heinze et al., 2017). In case of the 3D output, the simulation data is interpolated from the original grids (e.g., 156 m) to a 1 km grid, the 3D coarse data, and 300 m grid, the so-called HOPE data. The latter output has been created for the purpose of model evaluation with ground-based observations from the HD(CP)$^2$ Observational Prototype Experiment (HOPE) that took place near Jülich (Macke et al., 2017).

## 2.2   RRTMG

For radiative transfer simulations, ICON-LEM employs the rapid radiative transfer model (RRTM) for GCM applications (RRTMG) (Mlawer et al., 1997; Iacono et al., 2008). For the purpose of this investigation, an interface of the RRTMG for use with the Python programming language has been developed, which allows the offline calculation of the radiative fluxes using ICON-LEM outputs as basis.

RRTMG is a fast and accurate broadband radiative transfer model developed by the Atmospheric Environmental Inc. The

model employs the correlated-k approach for efficient fluxes and heating rates computations (Mlawer et al., 1997). Molecular absorption information for the k-distributions is taken from the line-by-line radiative transfer model (LBLRTM) (Clough et al., 2005). Fluxes and heating rates are derived for 14 bands in the SW and 16 bands in the LW. RRTMG considers major absorbing gases, i.e., water vapor, ozone, and carbon dioxide, but also minor absorbing species, i.e., methane, oxygen, nitrogen, and aerosols. Optical properties (optical thickness, single-scattering albedo, and asymmetry parameter) of liquid water clouds are

parameterized according to Hu and Stamnes (1993). Note that the RRTMG is a 1D plane-parallel radiative transfer model. For the representation of the sub-grid cloud variability, a Monte Carlo independent column approximation (McICA) method is used (Pincus et al., 2003). Multiple-scattering is considered employing a two-stream algorithm (Oreopoulos and Barker, 2006).

RRTMG provides the SW and LW radiative fluxes for both the upward ($F^{\uparrow}$) and downward ($F^{\downarrow}$) radiation. These two components can be combined to define the net flux ($F^{\mathrm{net}}$),

$$F^{\mathrm{net}} = F^{\downarrow} - F^{\uparrow}. \tag{2}$$



Accordingly, the cloud radiative effect (CRE) is defined as the difference between the cloudy and clear sky net radiative fluxes,

$$\mathrm{CRE} = F_{\mathrm{cloudy}}^{\mathrm{net}} - F_{\mathrm{clear}}^{\mathrm{net}}. \tag{3}$$

The CRE can be computed for the LW, SW, or the net CRE, defined by the sum of the SW and LW radiation.

### 2.3 Case days

5 In this study, the 3D HOPE data has been used and a set of 6 days of simulations have been considered, including: 24–25 April 2013, 5 May 2013, 29 July 2014, 14 August 2014, and 3 June 2016. Only a limited subset of variables is stored including the specific humidity, cloud water, ice, rain and snow mixing ratio, wind, vertical velocity, temperature, pressure, cloud cover, and turbulent diffusion coefficient for heat. These days have been selected from the total set of available case days by the presence of suitable liquid water cloud fields and no known bugs in the used model version, which affect the representation of low-level 10 clouds.

### 2.4 Column selection

In order to investigate the characteristics of liquid water clouds in ICON-LEM, only idealized cloud profiles (i.e., stratiform and cumulus) are considered, corresponding to single-layer non-drizzling clouds. The selection of such cloudy columns has been conducted according to requiring the following threshold criteria:

15 – For each cloudy layer, a liquid water content of $q_{\mathrm{L}} > 0.01\,\mathrm{g\,m^{-3}}$ and a liquid water path ($Q_{\mathrm{L}}$) larger than $20\,\mathrm{g\,m^{-2}}$.

– No occurrence of rain/drizzle; $Z_{\mathrm{max}} < -15\,\mathrm{dBZ}$, denoting the maximum radar reflectivity (see Eq. 6) within the cloud profile (Rémillard et al., 2013; Merk et al., 2016).

– A cloud geometrical extent ($H$) larger than $100\,\mathrm{m}$ (at least two subsequent model layers).

– Clouds located between $300\,\mathrm{m}$ and $4000\,\mathrm{m}$.

20 – No vertical gaps are allowed.

– Mixed-phase clouds are excluded. The ice water content for the first $4000\,\mathrm{m}$ must be zero.

– Superadiabatic clouds have been excluded.

The cloud bottom height (CBH) and cloud top height (CTH) are determined by the bottom and top of the lowermost and uppermost layers for the aforementioned ideal low-level clouds, respectively.





## 2.5 Cloud property diagnostics

The model provides in the output the droplet number concentration and liquid water content for each model layer representing the zeroth and the first moments of the mass size distribution (MSD, see Eq. 1). Following Petty and Huang (2011), the mass size distribution is transformed into a droplet size distribution (DSD). For details on the derivation of the moments of DSD and

the cloud microphysical properties, the reader is referred to Appendix A.

Following Hansen and Travis (1974), the effective radius, $r_{\mathrm{eff}}$, is defined as the ratio of the third to the second moments of the DSD,

$$r_{\mathrm{eff}} = \frac{1}{2} \frac{\int_0^\infty n(D)(D)^3 \, \mathrm{d}D}{\int_0^\infty n(D)(D)^2 \, \mathrm{d}D}. \tag{4}$$

The division by 2 is carried out for diameter-to-radius conversion. The effective radius is linked to the volume-equivalent radius

($r_{\mathrm{V}}$) by the $k_2$ factor, which depends only on the effective variance ($\upsilon$) of the droplet size distribution,

$$k_2 = \frac{r_{\mathrm{V}}^3}{r_{\mathrm{eff}}^3} = (1 - \upsilon)(1 - 2\upsilon). \tag{5}$$

For ICON-LEM, the effective variance of the reconstructed Gamma DSD is $\upsilon = 0.052$, corresponding to $k_2 = 0.849$. Typical values of $k_2$ reported in the literature vary between 0.5 and 1 (e.g., Brenguier et al., 2000; Zeng et al., 2014; Merk et al., 2016). Furthermore, the radar reflectivity is defined as the sixth moment of the size distribution,

$$Z = \int_0^\infty n(D)(D)^6 \, \mathrm{d}D. \tag{6}$$

Note, that in ICON-LEM, the droplet number concentration varies with height, but the width of the DSD is assumed invariant.

## 2.6 Cloud models

### 2.6.1 Vertically homogeneous cloud model

A widely used assumption for passive satellite and ground-based retrievals is the vertically homogenous cloud scheme. Ac-

cordingly, a vertically homogeneous DSD is assumed, meaning vertically constant microphysical properties. It follows that the cloud liquid water path is given by,

$$Q_{\mathrm{L}} = \frac{2}{3} \rho_{\mathrm{w}} \cdot \tau \cdot r_{\mathrm{eff}}, \tag{7}$$

describing a positive linear relationship between $Q_{\mathrm{L}}$ and both the cloud optical thickness ($\tau$) and effective radius ($r_{\mathrm{eff}}$). Here, $\rho_{\mathrm{w}}$ stands for the water density. Assuming a vertically constant cloud droplet number concentration additionally implies that

the cloud geometric extent depends linearly on the cloud water path for a fixed effective radius.



### 2.6.2 Sub-adiabatic cloud model

The sub-adiabatic cloud scheme describes the evolution of a convective closed parcel of moist air. According to Albrecht et al. (1990), the liquid water content ($q_L$) of such an air parcel increases linearly with height,

$$q_L(z) = f_{ad} \cdot \Gamma_{ad}(T(z), P(z)) \cdot z, \tag{8}$$

where $\Gamma_{ad}$ is the adiabatic increase of the liquid water content (Bennartz, 2007), $z$ is the height over the cloud base, $f_{ad}$ denotes the sub-adiabatic fraction, $T$ is the temperature, and $P$ is the pressure. $f_{ad}$ describes the deviation from the linear increase with height of $q_L$ caused by entrainment of dry air resulting in evaporation and $f_{ad} < 1$ (sub-adiabaticity). In case of a pure adiabatic cloud, $f_{ad} = 1$ and Eq. (8) yields to the adiabatic liquid water content ($q_{L,ad}$). For low-level liquid water clouds, typical values of $f_{ad}$ found in the literature are in the range of 0.3 to 0.9 (Boers et al., 2006). An alternative definition for the liquid water content accounting for the depletion of the liquid water content due to entrainment, precipitation, and freezing drops, is described by,

$$q_L = q_{L,ad}[1.239 - 0.145 \cdot \ln(z)], \tag{9}$$

following a modified sub-adiabatic profile (Karstens et al., 1994; Foth and Pospichal, 2017).

$\Gamma_{ad}$ depends on temperature (weak function of pressure) following the first law of thermodynamics and the Clausius–Clapeyron. For low-level clouds, its values vary slightly ($\sim 20\,\%$) and for most studies are assumed constant (e.g., Albrecht et al., 1990; Boers et al., 2006) or are calculated from cloud bottom temperature and pressure (e.g., Merk et al., 2016) or cloud top information (e.g., Zeng et al., 2014). For this study, an average value of $\Gamma_{ad}$ between cloud bottom and cloud top has been used.

Integrating the liquid water content between cloud base height and cloud top height, the cloud liquid water path is obtained,

$$Q_L = \int_{CBH}^{CTH} q_L(z)\,\mathrm{d}z = \frac{1}{2} f_{ad} \cdot \Gamma_{ad} \cdot H^2. \tag{10}$$

Hereby, $H$ denotes the cloud geometrical extent. Note that the ratio of the sub-adiabatic liquid water path to the equivalent vertically homogeneous one yields a factor of $5/6$. Dividing $Q_L$ by its adiabatic value (inserting $f_{ad} = 1$ into Eq. 10), the sub-adiabatic fraction can be computed,

$$f_{ad} = \frac{Q_L}{Q_{L,ad}}. \tag{11}$$

For low-level liquid water clouds, the droplet number concentration ($N_d$) depends on the availability of cloud condensation nuclei (CCN) that could get activated at cloud base (Bennartz, 2007). Considering the adiabatic increase of the liquid water





content, it follows that at any given height, $q_L$ is distributed over the activated CCN (per unit volume). Consequently, there is no dependency of the mean volume radius $r_V$ on the shape of the droplet size distribution, but only on $N_d$ and $q_L$,

$$r_V = \left(\frac{3q_L}{4\pi \cdot \rho_w \cdot N_d}\right)^{\frac{1}{3}}.$$
(12)

Combining Eqs. 5 and 12, the effective radius for the uppermost cloud layer can be written in terms of the liquid water path, the droplet number concentration, and the adiabatic fraction,

$$r_{\text{eff}}(Q_L, f_{\text{ad}}, N_d) = (18 f_{\text{ad}} \cdot \Gamma_{\text{ad}} \cdot Q_L)^{\frac{1}{6}} \left(4\pi \rho_w \cdot k_2 \cdot N_d\right)^{-\frac{1}{3}}.$$
(13)

In the geometric optics regime, the extinction coefficient, $b_{\text{ext}}$, can be written as a function of the liquid water content and the effective radius. Consequently, the cloud optical thickness can be computed by integrating $b_{\text{ext}}$ over the cloud geometrical extent, i.e., from cloud base height to cloud top height,

$$\tau = \int_{\text{CBH}}^{\text{CTH}} b_{\text{ext}}(z)\,\mathrm{d}z = \int_{\text{CBH}}^{\text{CTH}} \frac{3}{2\rho_w} \frac{q_L(z)}{r_{\text{eff}}(z)}\,\mathrm{d}z.$$
(14)

Alternatively, substituting $r_{\text{eff}}$ from Eq. (13) in Eq. (14), the cloud optical thickness is given by,

$$\tau(Q_L, f_{\text{ad}}, N_d) = \frac{9}{5}\left(4\pi k_2 \cdot N_d\right)^{\frac{1}{3}}\left(18\rho_w^4 \cdot f_{\text{ad}} \cdot \Gamma_{\text{ad}}\right)^{\frac{1}{6}} Q_L^{\frac{5}{6}}.$$
(15)

## 3   Cloud characteristics

### 3.1   General features

Table 1 lists the statistics of the cloud properties for all the case days individually and on average as simulated from ICON-LEM, while Fig. 1 illustrates the corresponding histograms in case of 3 June 2016 and the average over all days. Throughout this study, a special emphasis is given on 3 June 2016 cause it approximates best the mean properties over all the case days considered. Note that for the droplet number concentration and the effective radius, results are presented as follows:

– droplet number concentration weighted over the cloud geometrical extent, given by,

$$N_{\text{int}} = \frac{1}{H} \int_{\text{CBH}}^{\text{CTH}} N_d(z) \cdot \mathrm{d}z,$$
(16)

– effective radius weighted over the extinction coefficient at each layer,

$$r_{\text{int}} = \frac{1}{\tau} \int_{\text{CTH}}^{\text{CBH}} b_{\text{ext}}(z) \cdot r_{\text{eff}}(z) \cdot \mathrm{d}z.$$
(17)





**Table 1.** Statistics of cloud properties of low-level clouds for all the case days individually and on average as simulated from ICON-LEM. For the fraction of clouds two values are presented: values in brackets denote the fraction of selected clouds (FC) according to the column selection (see Sect. 2.4), while values outside brackets stand for the actual cloud fraction (CF) in terms of the following threshold for the liquid water path, $Q_L > 1\,\mathrm{g\,m^{-2}}$.

| Days | $n$ [-] | $Q_L$ [g m$^{-2}$] | $\tau$ [-] | CBH [m] | CTH [m] | $H$ [m] | $N_{\mathrm{int}}$ [cm$^{-3}$] | $r_{\mathrm{int}}$ [$\mu$m] | $f_{\mathrm{ad}}$ [-] | CF (FC) [%] |
|---|---|---|---|---|---|---|---|---|---|---|
| 24 April 2013 | 5822 | $41.9 \pm 20.7$ | $14.9 \pm 6.4$ | $641 \pm 163$ | $907 \pm 166$ | $266 \pm 56$ | $686 \pm 164$ | $4.1 \pm 0.4$ | $0.59 \pm 0.19$ | 1.75 (0.36) |
| 25 April 2013 | 29543 | $159.1 \pm 65.5$ | $37.4 \pm 43.8$ | $1721 \pm 285$ | $2262 \pm 323$ | $541 \pm 273$ | $380 \pm 154$ | $5.5 \pm 1.1$ | $0.47 \pm 0.21$ | 5.18 (1.83) |
| 5 May 2013 | 9465 | $60.2 \pm 48.8$ | $20.0 \pm 12.7$ | $1238 \pm 279$ | $1630 \pm 334$ | $391 \pm 127$ | $576 \pm 187$ | $4.2 \pm 0.6$ | $0.46 \pm 0.19$ | 2.57 (0.59) |
| 29 July 2014 | 48661 | $156.3 \pm 236.3$ | $39.3 \pm 48.8$ | $1063 \pm 601$ | $1599 \pm 662$ | $535 \pm 303$ | $464 \pm 195$ | $5.2 \pm 1.2$ | $0.40 \pm 0.19$ | 7.92 (3.02) |
| 14 August 2014 | 35105 | $114.3 \pm 192.7$ | $32.1 \pm 41.8$ | $779 \pm 533$ | $1214 \pm 625$ | $435 \pm 248$ | $612 \pm 229$ | $4.6 \pm 1.0$ | $0.48 \pm 0.19$ | 5.79 (2.18) |
| 3 June 2016 | 32768 | $116.0 \pm 152.0$ | $28.6 \pm 33.0$ | $1361 \pm 874$ | $1851 \pm 926$ | $491 \pm 241$ | $388 \pm 262$ | $5.7 \pm 1.4$ | $0.45 \pm 0.21$ | 17.2 (2.04) |
| All days | 161364 | $129.7 \pm 199.8$ | $33.2 \pm 41.5$ | $1177 \pm 675$ | $1644 \pm 746$ | $487 \pm 268$ | $480 \pm 232$ | $5.1 \pm 1.2$ | $0.45 \pm 0.21$ | 6.73 (1.67) |

It can be shown that the latter equation reduces to Eq. (7), which implies that the calculated effective radius corresponds to that of a vertically homogeneous cloud with identical liquid water path and optical thickness. The different cloud properties are characterized by a large variability from day to day, but even within the same day driven by entrainment processes. In addition, the differences are also subject to the sample size for each day depending on the column selection filter that applied

to ICON-LEM output. Recall here that a cloudy column is taken under consideration when $q_L > 0.01\,\mathrm{g\,m^{-3}}$ for each cloud model level while the liquid water path for the entire column should be larger than $20\,\mathrm{g\,m^{-2}}$. Subsequently, the fraction of clouds (FC) selected in this study is quite low (FC $< 3\,\%$). Alternatively, if only a liquid water path filter is applied to the data, defining as cloudy the columns with $Q_L$ larger than $1\,\mathrm{g\,m^{-2}}$, the actual cloud fraction (CF) is obtained. The rather large value of the CF found for 3 June 2016 is associated with very low (with $100 < \mathrm{CBH} < 200\,\mathrm{m}$) overcast cloudy conditions in the

early hours.

Looking at the mean histograms of CTH and CBH, one can identify multimodal distributions. Note here that, in this study, all the low-level clouds are considered (i.e., cumuli-like, stratiform) increasing the variability of the different properties.

The double-moment microphysical scheme adopted in ICON-LEM is reflected on the histograms of the droplet number concentration. The mean distribution of $N_{\mathrm{int}}$ suggests a bimodal distribution with peaks centered around $200\,\mathrm{cm^{-3}}$ and $450\,\mathrm{cm^{-3}}$.

For 3 June 2016, the peak around $200\,\mathrm{cm^{-3}}$ is even more notable. Note here that this value is close to the fixed droplet number concentration profile suggested by single-moment microphysical schemes adopted by atmospheric models (e.g., ECHAM). On the contrary, for the 24–25 April 2013, only a single mode is clearly identified (not shown here), with a peak towards large $N_d$ values for the 24$^{\mathrm{th}}$ and small values for the 25$^{\mathrm{th}}$ centered around $686\,\mathrm{cm^{-3}}$ and $380\,\mathrm{cm^{-3}}$, respectively. A close relation between the effective radius and the droplet number concentration exist. On average, the larger the $N_{\mathrm{int}}$ the smaller the $r_{\mathrm{int}}$.

## 3.2 Vertical variability

Figure 2 shows a box-whisker plot of the droplet number concentration for 3 June 2016, describing the histograms of $N_d$ simulated for different model levels by the double moment scheme of ICON-LEM. For comparison, the red line shows the





climatology-based droplet number concentration profile adopted by ECHAM (Giorgetta et al., 2013). While above $2\,\mathrm{km}$ altitude, the modeled values match the climatology well, much larger median values up to $600\,\mathrm{cm^{-3}}$ are found in the boundary layer. Compared to satellite estimates of $N_\mathrm{d}$, these values seem excessively high (Quaas et al., 2006; Grosvenor et al., 2018). Furthermore, in situ observations suggest higher values of $N_\mathrm{d}$ closer to those simulated by ICON-LEM, but are affected by

5    large instrumental uncertainties (Grosvenor et al., 2018). Hence, efforts should be undertaken to validate the cloud droplet number concentrations predicted by the double-moment scheme.

Figure 3 depicts the mean profiles of $q_\mathrm{L}$ and $N_\mathrm{d}$ normalized over the cloud geometrical extent (from CBH to CTH) for 3 June 2016. The ICON-LEM simulated liquid water profile follows a linear increase from cloud bottom to around $60\,\%$ of the cloud height in agreement with the adiabatic cloud model. Thereafter, the liquid water content decreases towards the cloud top

10    due to evaporation induced by entrainment of dry air mass from cloud top. Furthermore, the mean profile of the droplet number concentration is found roughly constant at verticals depths between 20 and $75\,\%$ of $H$ ($\sim 400\,\mathrm{cm^{-3}}$) and decreases towards the cloud top at values $\sim 150\,\mathrm{cm^{-3}}$ characterized by a large variability.

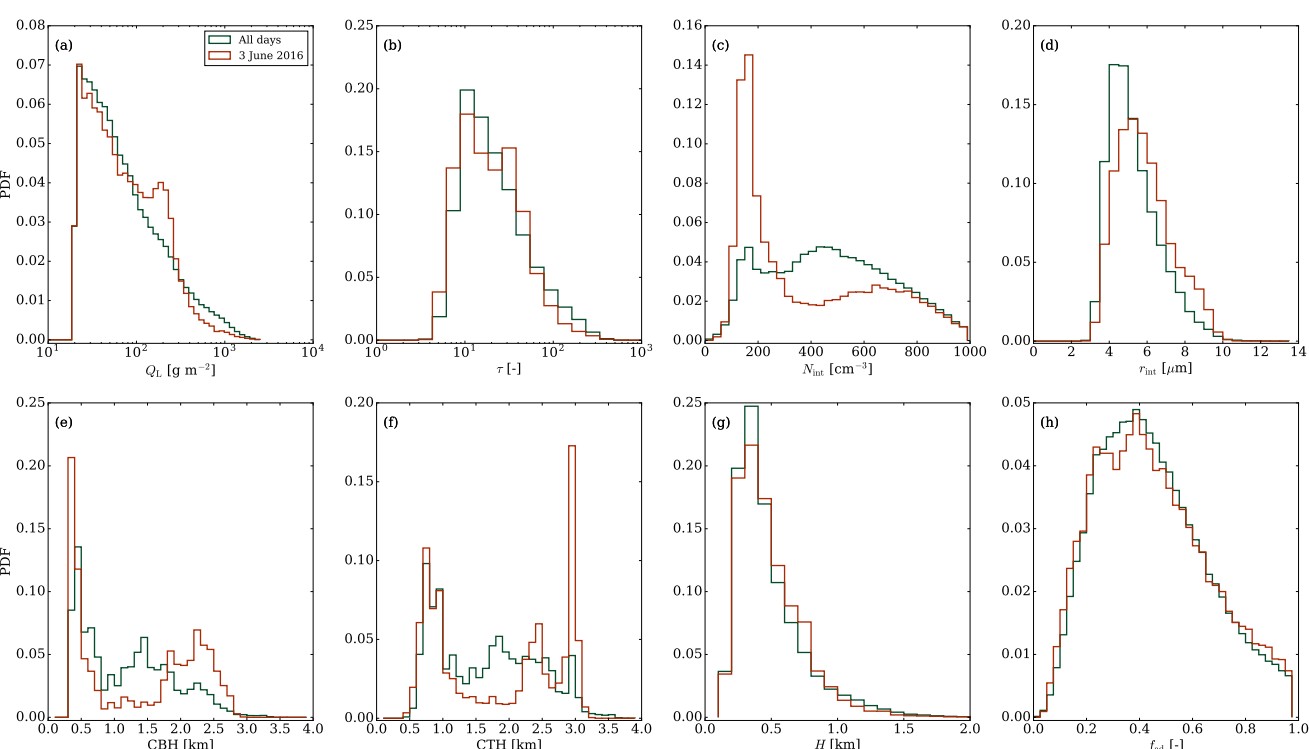

**Figure 1.** Histograms of cloud properties of low-level clouds in case of 3 June 2016 (red) and on average for all the case days (green) as simulated from ICON-LEM: (a) $Q_\mathrm{L}$, (b) $\tau$, (c) $N_\mathrm{int}$, (d) $r_\mathrm{int}$, (e) CBH, (f) CTH, (g) $H$, and $f_\mathrm{ad}$.

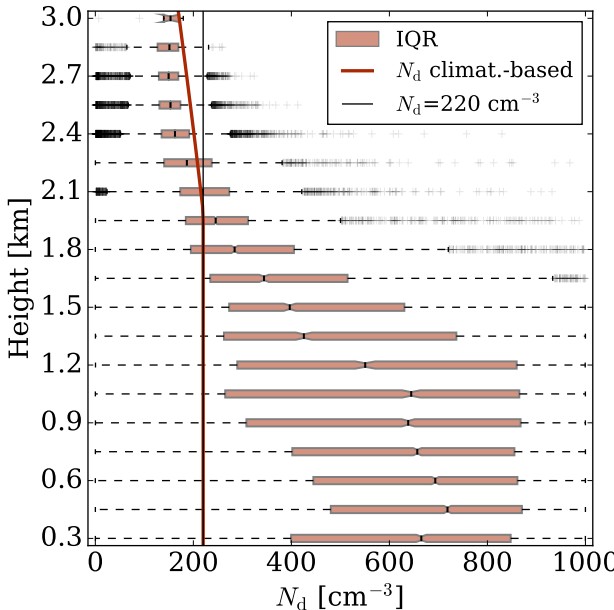

**Figure 2.** Box-whisker plot of the droplet number concentration for 3 June 2016, describing the histograms of $N_d$ simulated for different model levels by the double moment scheme of ICON-LEM. Boxes illustrate interquartile range (IQR), dark red line denotes the vertical $N_d$ profile in case of the droplet number concentration employed in coarse climate models (climat.-based) and the thin black line demonstrates the constant $N_d$ profile of $220 \, \text{cm}^{-3}$.

### 3.3 Adiabaticity of liquid water clouds

Following the sub-adiabatic cloud model, higher values of the liquid water path are linked with geometrically thicker clouds (see Eq. 10). For all the days, the distribution of the cloud geometrical extent follows a similar pattern, except for 24 April 2013 and 5 May 2013. For the latter two days, only optically thinner clouds are simulated as compared to the rest days, with $\tau$
5    values of 14.9 and 20, respectively. However, this could also be subject to the very small sample size as compared to the other simulated days. The highest mean value of the sub-adiabatic fraction is found for the 24 April 2013, whereby only optically and geometrically thin clouds are simulated located at the lowermost altitudes (mean CTH of 907 m). One could expect the same findings for the 5 May 2013, but the smaller values of $f_{ad}$ are partly associated with the higher values of $H$ together with their vertical location where entrainment processes can be more pronounced. The lowest mean values of $f_{ad}$ are found
10   for 29 July 2014 reflected by the high frequency of occurrence of larger values of the cloud geometrical extent. Overall, the statistics of $f_{ad}$ for the six days under investigation (161364 liquid water cloudy columns) over Germany introduces a mean value of about $f_{ad} = 0.45$ (see Table 1) while the interquartile range (IQR) is [0.29, 0.59]. There is a wide range of values of $f_{ad}$ from nearly 0 to 1. The latter is in agreement with the findings of Boers et al. (2006); Merk et al. (2016). Especially, Merk





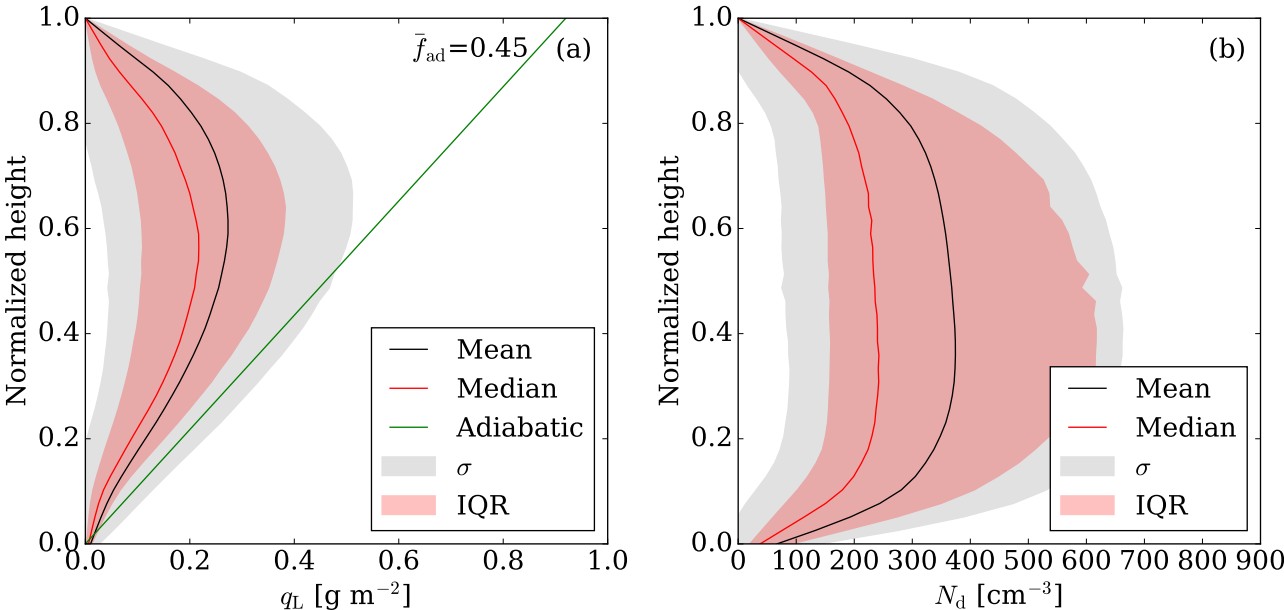

**Figure 3.** ICON-LEM simulated mean (a) $q_L$ and (b) $N_d$ profiles for 3 June 2016. Profiles are normalized over height from the CBH to the CTH. Black lines denote the mean, red solid lines the median, gray shaded areas the standard deviation, red shaded areas the interquartile range (IQR), and the green solid line outline the mean adiabatic $q_L$ profile characterized by a mean adiabatic fraction ($\bar{f}_{ad}$) of 0.45.

**Table 2.** Ordinary least squares regression results: Regressor coefficients ($a$), $Y$-intercept ($a_0$), squared correlations ($R^2$), and root-mean-square error (RMSE). Theoretical (Th.) values according to the sub-adiabatic model are also included.

| | | $Y = a_0 + a_1 \cdot x_1 + ... + a_n \cdot x_n$ | | | | |
|---|---|---|---|---|---|---|
| $Y$ | $a_0$ | $a_1 \cdot \ln(Q_L)$ | $a_2 \cdot \ln(f_{ad})$ | $a_3 \cdot \ln(N_{int})$ | $R^2$ | RMSE |
| $Y_1$ | $-0.557 \pm 0.0020$ | $0.849 \pm 0.0004$ | - | - | 0.957 | 0.175 |
| $Y_2$ | $-2.037 \pm 0.0019$ | $0.808 \pm 0.0002$ | - | $0.274 \pm 0.0003$ | 0.992 | 0.075 |
| $Y_3$ | $-0.665 \pm 0.0024$ | $0.860 \pm 0.0005$ | $0.065 \pm 0.0009$ | - | 0.959 | 0.172 |
| $Y_4$ | $-2.437 \pm 0.0008$ | $0.830 \pm 0.0001$ | $0.147 \pm 0.0001$ | $0.303 \pm 0.0001$ | 0.999 | 0.027 |
| Th. | - | $a_1 = 0.833$ | $a_2 = 0.167$ | $a_3 = 0.333$ | - | - |

et al. (2016) derived the $f_{ad}$ from ground-based observations over Germany and reported a mean value of 0.45 for the period 2012–2015, with a IQR of [0.29, 0.61]; Boers et al. (2006) reported $f_{ad}$ values within [0.3, 0.9].





### 3.3.1 Cloud optical thickness

One of the fundamental cloud properties describing the SW radiative effect is the cloud optical thickness. Thus, we focus on its derivation and its dependencies.

On a logarithmic scale, Eq. (15) suggests that $\tau$ is a linear function of $Q_L$, $f_{ad}$, and $N_d$ and it can be seen as a linear regression model. Here, the droplet number concentration weighted over the cloud geometrical extent ($N_{int}$) is used. An advantage of the logarithmic scale is that the variance of the cloud optical thickness can be decomposed into the contributions from each of the regressors ($Q_L$, $f_{ad}$, and $N_{int}$). This enables us to attribute the relative importance of the regressors in explaining the variance in $\tau$. In our framework, we employed the ordinary least squares (OLS) regression method. This method finds the projection direction for which $Q_L$, $f_{ad}$, and $N_{int}$ are maximally correlated with $\tau$ and provides the values of the coefficients that minimize the error in the prediction of $\tau$. Results are compiled in Table 2.

Firstly, we focus on the relative importance of $Q_L$ in $\tau$. Model $Y_1(Q_L)$ suggests that the liquid water path explains 95.7 % of the variance in cloud optical thickness and they follow an excellent linear relationship (see Fig. B1 in Appendix B) with a $5/6$ fit ($\alpha = 0.8489$) and a root mean square error (RMSE) of 0.1751. In agreement with the sub-adiabatic model, $\tau$ is proportional to $Q_L^{5/6}$ and not to $Q_L$ as suggested by the vertically homogeneous model; otherwise, a value of $\alpha = 1$ would be expected. Comparing the models $Y_2(Q_L, N_{int})$ and $Y_3(Q_L, f_{ad})$, $Y_2$ has a higher $R^2$ value (0.9921 % compared to 0.9585 %), a lower RMSE (0.0750 compared to 0.1723), while the regression coefficients are much closer to the sub-adiabatic theory.

All in all, the liquid water path is able to explain 95.71 % of the variance in cloud optical thickness, while the droplet number concentration and the sub-adiabatic fraction additionally contribute 3.5 % and 0.14 % to the variance, respectively.

Variability caused by $\Gamma_{ad}$ is insignificant and, thus, is not shown here. This is confirmed by model $Y_4(Q_L, f_{ad}, N_{int})$, which, even though it excludes $\Gamma_{ad}$, explains 99.9 % of the variance in cloud optical thickness.

## 4 Principal component analysis

To identify the minimum set of parameters for the representation of low-level clouds towards the computation of the CREs, the dominating modes of variability among the different cloud properties have been investigated. Cloud properties from all the case days have been considered. $\Gamma_{ad}$ is not a cloud property, but since it is considered by the sub-adiabatic model, we decided to include it in the analysis. Towards this direction, one should first map the correlation of the different properties. Figure 4 identifies groups of variables that tend to covary together. The first group comprises $\tau$, $Q_L$, and $H$ that are strongly positively correlated with one another ($R^2 > 0.83$), while in the second group, CTH, CBH are positively correlated ($R^2 > 0.93$) albeit inversely correlated with $\Gamma_{ad}$ ($R^2 < -0.88$). Alternatively, these two groups could be partly noted as the SW and LW (excluding $\Gamma_{ad}$) properties, respectively. Last but not least, only a weak to mediocre correlation was found between $r_{int}$, $N_{int}$, $f_{ad}$ and the other properties.

A principal component analysis (PCA) is applied to reveal systematic co-variations among the cloud properties, reducing the degrees of freedom, while preserving the maximum amount of information towards redundancy. Since our aim is to retain as few degrees of freedom as possible, the first step is to estimate the optimized number of components needed. As a primary



**Table 3.** Pearson correlations between the logarithm of the cloud properties and the principal (PC) and rotational components (RC). Degree of correlation (absolute values): (a) very weak: below 0.2, (b) weak: [0.2, 0.4), (c) moderate: [0.40, 0.6), (d) strong: [0.6, 0.8), and (e) very strong [0.8, 1.0].

| Properties | PC-1 | RC-1 | PC-2 | RC-2 | PC-3 | RC-3 | PC-4 | RC-4 |
|---|---|---|---|---|---|---|---|---|
| CBH | 0.78 | 0.97 | −0.53 | 0.03 | −0.08 | 0.00 | −0.25 | 0.20 |
| CTH | 0.92 | 0.92 | −0.29 | −0.28 | 0.07 | 0.08 | −0.20 | 0.24 |
| $\Gamma_{\text{ad}}$ | −0.74 | −0.90 | 0.49 | −0.01 | 0.16 | 0.07 | 0.27 | −0.18 |
| $\tau$ | 0.45 | −0.06 | 0.89 | −0.97 | 0.06 | −0.19 | −0.09 | −0.12 |
| $Q_{\text{L}}$ | 0.59 | 0.04 | 0.81 | −0.97 | 0.00 | −0.24 | 0.02 | 0.05 |
| $H$ | 0.66 | 0.18 | 0.57 | −0.94 | 0.48 | 0.29 | −0.01 | 0.09 |
| $f_{\text{ad}}$ | 0.10 | −0.10 | 0.34 | −0.10 | −0.94 | −0.99 | 0.00 | −0.03 |
| $N_{\text{int}}$ | −0.53 | −0.52 | 0.70 | −0.25 | −0.12 | −0.24 | −0.45 | −0.78 |
| $r_{\text{int}}$ | 0.86 | 0.38 | 0.16 | −0.54 | −0.22 | −0.31 | 0.43 | 0.68 |

solution, we used the same number of components as the original variables (nine in number) and we estimated the fraction of variance explained by each component. Figure 5 illustrates the resulting cummulative explained variance as a function of each principal component (PC). The cumulative explained variance suggests the use of four PCs in the logarithmic space (98 %), going from a nine-dimensional space to a four-dimensional space; the variance contributed by the fifth component is bellow

1.87 %. The interpretation of the principal components is based on finding which properties are mostly strong correlated with each component. Table 3 summarizes the quality of reduction in the squared correlations (Pearson) by comparing the residual correlations (PCs) to the logarithm of the original cloud properties. However, the PCs are hard to interpret since, although each new dimension is clearly dominated by some cloud properties, they are found moderately or strongly correlated with other properties. For example, PC-2, which explains 33.4 % of the total variance, is driven by $\tau$, $Q_{\text{L}}$, $N_{\text{int}}$, and $H$ and is substantially

correlated with CBH and $\Gamma_{\text{ad}}$.

     Subsequently, the so-called varimax rotation has been utilized in order to associate each cloud property to at most one principal component by maximizing the sum of the variances of the squared correlations between the cloud properties and the PCs (Stegmann et al., 2006). This results in the rotational components (RCs). RCs are also compiled in Table 3. Under those circumstances, the resulting squared correlations are either close to unity or zero, allowing only a few moderate correlations

and pointing to how each cloud property loads on each component, while preserving the overall number of components (see Fig. 5). Please note the differences between PCs and RCs. RC-2, responsible for 36 % of the variance in logarithmic space, is strongly correlated with three of the original variables. Considering the strong correlation found between $\tau$ and $Q_{\text{L}}$ (see Fig. 4) and their robust linear relation ($R^2 = 0.99$), they can be considered interchangeable. In the same direction are the findings for RC-1 and CBH, CTH, and $\Gamma_{\text{ad}}$, with an explained variance of about 34 %. The RC-3 and RC-4 are clearly a function of $f_{\text{ad}}$

and $N_{\text{int}}$, respectively, pointing to two clear degrees of freedom. Effective radius is the only property that shows a moderate importance in more than one RCs, namely RC-1, RC-3, and RC-4. $r_{\text{int}}$ could be substituted as a degree of freedom from a well





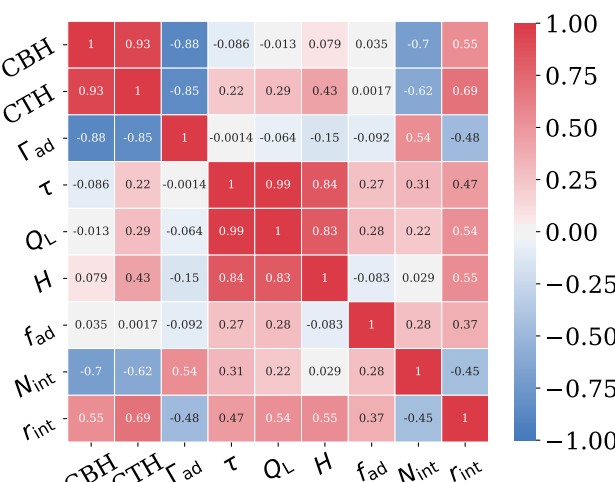

**Figure 4.** Correlation plot between all the properties (CBH, CTH, $\Gamma_{ad}$, $\tau$, $Q_L$, $H$, $f_{ad}$, $N_{int}$, and $r_{int}$).

defined DSD, with $N_{int}$ as a primary component and $k_2$. Note here that the first two components account for more than 69 % (in logarithmic scale) of the variance of the cloud properties with the first component related to those that dominate in the SW CRE while, the second component, with those that are of great importance in the LW CRE.

The aforementioned analysis points to the reduced set of parameters for the representation of low-level clouds towards the computation of the CREs: $N_{int}$, $Q_L$, $f_{ad}$, $H$, and one of the CTH or CBH.

## 5 Cloud radiative effects of low-level clouds

### 5.1 Radiative transfer simulations

The input for the radiative transfer simulations was constructed on the basis of ICON-LEM. In other words, temperature, pressure, and water vapour profiles, surface temperature and pressure, and the cloud's liquid water content and droplet number concentration are taken from the high-resolution model. For ozone, the profile of the US standard atmosphere is adopted (Anderson et al., 1986). Note here that the ICON-LEM profiles reach approximately 20 km altitude. We further extended the atmosphere up to 120 km height again using the US standard atmosphere. The carbon dioxide concentration was set to 399 ppm. Simulations have been conducted only for one day, 3 June 2016. Considering the focus of this work, the following assumptions have been made: constant values for the direct and a diffuse SW surface albedo for the ultraviolet/visible (0.05)





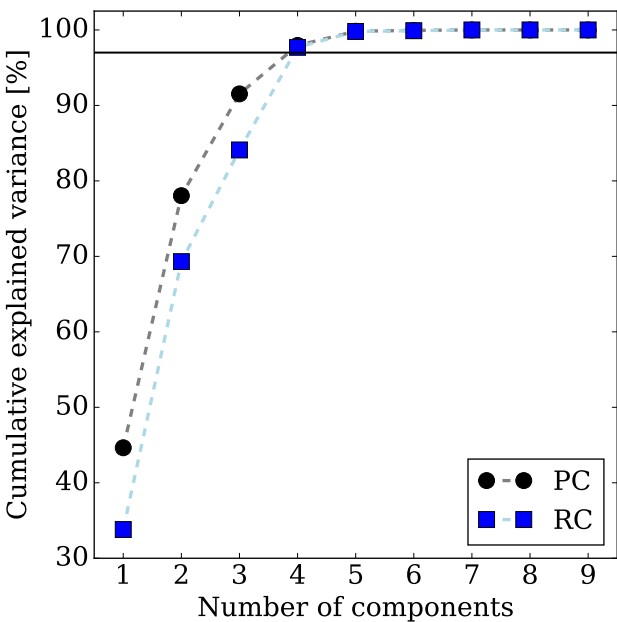

**Figure 5.** Explained variance by different components for both the principal component analysis (PC) and the rotational component analysis (RC).

and near-infrared (0.3) band and the cosine of the solar zenith angle (0.7); the effects of aerosols are neglected. Last but not least, maximum overlap of cloudy layers is assumed, since only idealized single-layer liquid water clouds are considered for this study.

### 5.1.1 Simulated scenarios

5 In order to estimate the effects of the bulk microphysical parameterizations and the vertical stratification of the cloud properties on the CREs, the double-moment scheme (ICON-LEM; hereafter reference simulation, Ref.) is confronted against the following scenarios: S1, single-moment scheme, whereby the droplet number concentration follows a fixed profile that varies according to pressure profile ($P$), sharing the same liquid water content profiles as in Ref.,

$$N_\mathrm{d}(P) = N_{\mathrm{d},1} + (N_{\mathrm{d},2} - N_{\mathrm{d},1}) \cdot \mathrm{e}^{f(P)}, \tag{18}$$

10 with,

$$f(P) = \min(8, P_\mathrm{b}/P)^2. \tag{19}$$





**Table 4.** Simulated scenarios. For scenarios S1–S4, four individual simulations have been conducted according to different values for the droplet number concentration profile.

| Scenarios | |
|---|---|
| Ref. | Double-moment scheme |
| S1 | Single-moment scheme |
| S2 | Vertically homogeneous model |
| S3 | Sub-adiabatic model |
| S4 | Modified sub-adiabatic model |
| a. $220\,\mathrm{cm^{-3}}$ | b. $N_{\mathrm{int}}$  c. $388\,\mathrm{cm^{-3}}$  d. 217 or $686\,\mathrm{cm^{-3}}$ |

Here, $N_{\mathrm{d,2}}$ is the droplet number concentration in the boundary layer, $N_{\mathrm{d,1}} = 50\,\mathrm{cm^{-3}}$ denotes the corresponding value in the free troposphere, and $P_{\mathrm{b}}$ is the boundary layer height (800 hPa) (Giorgetta et al., 2013). Three different scenarios are considered, where the liquid water path in preserved, but redistributed within the profile. In S2, a constant liquid water content profile is used with a fixed droplet number concentration representing the vertically homogeneous cloud model. Scenario S3 denotes the equivalent sub-adiabatic profile. Finally, following Karstens et al. (1994), Foth and Pospichal (2017), a modified sub-adiabatic profile is considered in S4 accounting for entrainment processes. For scenarios S1–S4, four individual simulations have been conducted according to the droplet number concentration profile:

- $N_{\mathrm{d}}$ following the climatology of coarse atmospheric models (e.g., ECHAM, Giorgetta et al., 2013), $220\,\mathrm{cm^{-3}}$.

- $N_{\mathrm{d}}$ weighted over $H$, $N_{\mathrm{int}}$.

- $N_{\mathrm{d}} = 388\,\mathrm{cm^{-3}}$, employing the mean $N_{\mathrm{int}}$ for 3 June 2016.

- $N_{\mathrm{d}}$ defined according to the two major clusters in the histogram of $N_{\mathrm{d}}$ for 3 June 2016 (see Fig 1). If $N_{\mathrm{int}}$ is below $388\,\mathrm{cm^{-3}}$, a value of $217\,\mathrm{cm^{-3}}$ is used, while, if $N_{\mathrm{int}}$ is larger than $388\,\mathrm{cm^{-3}}$, a value of $686\,\mathrm{cm^{-3}}$ is used defined as the mean values over the two clusters, respectively.

Note here that all scenarios share the same $Q_{\mathrm{L}}$ and $k_2$ parameter. The different scenarios are summarized in Table 4.

### 5.1.2 Modelled CREs

For the reference run, the mean and the standard deviation of the modeled CREs for the SW, LW, and NET (SW + LW) radiation are summarized in Table 5. The atmospheric cloud radiative effect (ATM) defined as the difference between CREs at the TOA and BOA is also included. Results are presented for 3 June 2016. Low-level clouds induce a strong negative SW CRE, driven by vigorous scattering, and a positive LW CRE, due to absorption of upward radiation, resulting in a net cooling effect. The warming of the atmosphere due to absorption of SW radiation ($\sim 37.3\,\mathrm{W\,m^{-2}}$) is recompensed by the atmospheric LW cooling ($\sim -44.1\,\mathrm{W\,m^{-2}}$), leading to a net cooling of the atmosphere ($\sim -6.84\,\mathrm{W\,m^{-2}}$). The net CRE is characterized by high variability depending on the distribution of the microphysical and optical cloud properties (see Sect. 5.1.3).





**Table 5.** Mean and standard deviation of modeled CREs ($\text{W m}^{-2}$) for the SW, LW, and NET (SW + LW) radiation for the reference simulation. Radiative transfer simulations have been conducted for 3 June 2016 corresponding to a Julian day number of 155. The cosine of the solar zenith angle is set to 0.7. ATM stands for the atmospheric cloud radiative effect defined as the difference between the CREs at the TOA and BOA.

| Ref. | $\text{CRE}_{\text{SW}}$ | $\text{CRE}_{\text{LW}}$ | $\text{CRE}_{\text{NET}}$ |
|---|---|---|---|
| TOA | $-331.1 \pm 77.5$ | $9.88 \pm 6.32$ | $-323.3 \pm 79.2$ |
| ATM | $37.3 \pm 13.9$ | $-44.1 \pm 12.5$ | $-6.84 \pm 10.1$ |
| BOA | $-370.4 \pm 89.7$ | $54.0 \pm 6.62$ | $-316.4 \pm 86.8$ |

Table 6 lists the mean CREs between the reference and the rest simulated scenarios for the SW radiation for both TOA and BOA. For the LW, all the scenarios are able to reproduce the reference CREs (see Table C1 in Appendix C); the mean CRE is below $\sim 0.5\,\text{W m}^{-2}$ with the vertically homogeneous run leading to the largest differences. Note here that the deviations in the CREs for the BOA and the TOA are of the same magnitude.

Overall, the single-moment radiative transfer simulations underestimate the SW CREs for both the TOA and BOA. Starting from S1a, the CREs in the single-moment run is $-39.5\,\text{W m}^{-2}$ less than the double-moment one, with a root mean square error (RMSE) up to $48.8\,\text{W m}^{-2}$. The latter differences are attributed to the very low droplet number climatology adopted by coarse climate models as compared to ICON-LEM. For a given liquid water path, the smaller the droplet number concentration the larger the resulting effective radius and, accordingly, the smaller the cloud reflectance. In other words, this can be seen as

the magnitude of the cloud albedo effect, the so-called first indirect effect (e.g., Twomey, 1977; Ackerman et al., 2000; Werner et al., 2014). For 3 June 2016, a mean value of $388 \pm 262\,\text{cm}^{-3}$ is found for the droplet number concentration and a fixed $N_{\text{d}}$ profile of $220\,\text{cm}^{-3}$ can only represent a fraction of the bimodal distribution of the droplet number concentration yielded from ICON-LEM (see also Fig. 1). A single-moment run with a more representative profile for the droplet number concentration approximates the SW CRE with more accuracy. By employing the mean $N_{\text{d}}$ (S1c), the differences in the CRE between the

single- and the double-moment runs are considerably smaller, but with quite large scatter; for the BOA (TOA), a RMSE of $31.2\,\text{W m}^{-2}$ ($31.5\text{W m}^{-2}$) and a Pearson correlation of 0.950 (0.928) is yielded. Furthermore, representing the two modes in the histogram of the droplet number concentration (Fig. 1) with the corresponding mean values of each mode ($217\,\text{cm}^{-3}$ and $686\,\text{cm}^{-3}$, S1d), leads to very small differences (up to $6.84\,\text{W m}^{-2}$ with a RMSE up to $16.1\,\text{W m}^{-2}$ for both the BOA and TOA). The best scenario is found to be S1b, with a RMSE of $13.2\,\text{W m}^{-2}$ and a Pearson correlation of 0.995, but this is no

suprise considering the quite realistic representation of the droplet number concentration profile.

     Having preserved the liquid water path profile (but re-distributed, scenarios 2–4), one can regard the changes in the CREs to the vertical stratification of low-level clouds within ICON-LEM. Comparing the SW CREs yielded by the vertically homogeneous (S2) and the sub-adiabatic runs (S3 and S4), it follows that the shape of the liquid water content profile and, thus, the other cloud properties can be well represented by the sub-adiabatic model. For the simulations with the more representative

droplet number concentration profiles (b and d), differences in CREs are more pronounced for the vertically homogeneous (of about $-6.52\,\text{W m}^{-2}$ with a RMSE of $10.4\,\text{W m}^{-2}$ for b and $-9.31\,\text{W m}^{-2}$ with a RMSE of $19.4\,\text{W m}^{-2}$ for d) simulation





**Table 6.** Mean CRE ($\mathrm{W\,m^{-2}}$). Results are given as differences between the new scenario minus the reference simulation ($\Delta$). The root mean square error (RMSE) in $\mathrm{W\,m^{-2}}$ and the Pearson (Pears.) correlation between the new scenarios and the reference simulation are also given.

| Scen. | $\mathrm{CRE_{SW,B}}$ | | | $\mathrm{CRE_{SW,T}}$ | | |
|-------|-------|-------|-------|-------|-------|-------|
|       | $\Delta$ | RMSE | Pears. | $\Delta$ | RMSE | Pears. |
| S2  | $-39.5$ | 48.8 | 0.951 | $-39.5$ | 48.6 | 0.934 |
| S2b | $-9.07$ | 13.2 | 0.996 | $-8.59$ | 12.9 | 0.994 |
| S2c | $-12.8$ | 31.2 | 0.952 | $-12.3$ | 31.5 | 0.930 |
| S2d | $-6.84$ | 16.1 | 0.989 | $-6.36$ | 15.9 | 0.985 |
| S3a | $-25.2$ | 40.6 | 0.938 | $-25.6$ | 41.2 | 0.911 |
| S3b | 6.52 | 10.4 | 0.997 | 6.64 | 10.2 | 0.996 |
| S3c | 3.15 | 33.2 | 0.937 | 3.38 | 34.7 | 0.901 |
| S3d | 9.31 | 19.4 | 0.985 | 9.46 | 19.8 | 0.979 |
| S4a | $-29.3$ | 42.9 | 0.940 | $-30.6$ | 44.1 | 0.914 |
| S4b | 1.10 | 8.36 | 0.996 | 0.38 | 7.94 | 0.995 |
| S4c | $-1.24$ | 32.1 | 0.941 | $-1.97$ | 33.7 | 0.906 |
| S4d | 3.64 | 16.5 | 0.987 | 2.96 | 16.8 | 0.981 |
| S5a | $-27.4$ | 41.7 | 0.939 | $-28.4$ | 42.8 | 0.913 |
| S5b | 4.14 | 8.83 | 0.997 | 3.58 | 8.43 | 0.996 |
| S5c | 1.05 | 32.6 | 0.939 | 0.57 | 34.2 | 0.903 |
| S5d | 6.73 | 17.6 | 0.987 | 6.22 | 17.9 | 0.981 |

as compared to the sub-adiabatic simulations (of about $-1.10\,\mathrm{W\,m^{-2}}$ with a RMSE of $8.36\,\mathrm{W\,m^{-2}}$ for b and $-3.64\,\mathrm{W\,m^{-2}}$ with a RMSE of $16.5\,\mathrm{W\,m^{-2}}$ for d). The dependency of the latter deviations on the different droplet number concentration profiles follows the same pattern as for the single- vs double-moment schemes. For instance, in case of the adiabatic scenarios and, going from the least to the most accurate ones, errors (in terms of the RMSE) up to $42.9\,\mathrm{W\,m^{-2}}$ for a, $32.6\,\mathrm{W\,m^{-2}}$ for c,

5 $17.6\,\mathrm{W\,m^{-2}}$ for d, and $8.83\,\mathrm{W\,m^{-2}}$ for b are found. This is in agreement with our findings in Sect. 3.2 and Sect. 3.3. Between the two sub-adiabatic runs (S3 and S4), they both approximate the CREs of the reference simulation with very high accuracy; slightly larger scatter is found for S4 as compared to S3 (e.g., for b, a RMSE of $8.83\,\mathrm{W\,m^{-2}}$ and $8.36\,\mathrm{W\,m^{-2}}$, respectively). For an illustration of the excellent linear correlation between the reference simulation and S4d by means of a bivariate kernel density (BKD) plot, the reader is referred to Fig. B2 in Appendix B. One can see that the CREs computed by these scenarios

10 are in a very good agreement almost everywhere except towards larger values of the CREs in case of the SW radiation.

Note here that discrepancies between the scenarios might exist subject to limitations of the radiative transfer model, i.e., RRTMG is able to derive the radiative fluxes only for effective radius between $2.5\,\mu\mathrm{m}$ and $60\,\mu\mathrm{m}$. Scenarios associated with very low (high) values of the droplet number concentration might result in very high (low) values of the effective radius and, thus, might not fulfill the above valid range.



**Table 7.** Spearman (Spear.) and Pearson (Pears.) correlations between the cloud radiative effects for the reference simulation (Ref.) and the cloud properties.

| Properties | $CRE_{SW,B}$ | | $CRE_{SW,T}$ | | $CRE_{LW,B}$ | | $CRE_{LW,T}$ | |
|---|---|---|---|---|---|---|---|---|
| | Spear. | Pears. | Spear. | Pears. | Spear. | Pears. | Spear. | Pears. |
| $Q_L$ | $-0.953$ | $-0.639$ | $-0.962$ | $-0.649$ | $0.220$ | $0.065$ | $0.106$ | $0.068$ |
| $\tau$ | $-0.998$ | $-0.708$ | $-0.996$ | $-0.714$ | $0.340$ | $0.132$ | $-0.072$ | $-0.005$ |
| $N_{int}$ | $-0.680$ | $-0.685$ | $-0.649$ | $-0.653$ | $0.519$ | $0.646$ | $-0.512$ | $-0.671$ |
| $r_{int}$ | $-0.177$ | $-0.149$ | $-0.211$ | $-0.186$ | $-0.401$ | $-0.486$ | $0.694$ | $0.700$ |
| CBH | $0.390$ | $0.497$ | $0.335$ | $0.435$ | $-0.819$ | $-0.906$ | $0.759$ | $0.941$ |
| CTH | $0.057$ | $0.294$ | $-0.005$ | $0.226$ | $-0.788$ | $-0.900$ | $0.897$ | $0.975$ |
| $H$ | $-0.760$ | $-0.696$ | $-0.784$ | $-0.718$ | $-0.014$ | $-0.003$ | $0.248$ | $0.146$ |
| $f_{ad}$ | $-0.299$ | $-0.267$ | $-0.291$ | $-0.257$ | $0.124$ | $0.101$ | $0.068$ | $0.018$ |

### 5.1.3 Impact of the cloud properties on the CREs

For a better assessment of the impact of the different cloud properties on both the SW and LW CREs their correlations have been investigated (in case of Ref.). Table 7 summarizes the corresponding Spearman (monotonic relation) and Pearson (linear relation) correlations while Fig. 6 and Fig. 7 illustrate the resulting bivariate kernel density between the cloud radiative effects and the cloud properties that are essential to describe the SW and LW radiation, respectively.

In the SW radiation, there is an excellent monotonic relation between the CREs and $\tau$, $Q_L$, and $H$ for both BOA and TOA, with spearman correlations higher than $-0.996$, $0.953$, and $-0.76$, respectively (see Table 7 and Fig. 6), following the second rotational component (RC-2, see Table 3). In particular, the SW CREs for both BOA and TOA increase monotonically with the liquid water path. The latter monotonic relation that is found stronger for lower values of the liquid water path saturates at $Q_L > 300\,\mathrm{g\,m^{-2}}$. In the same direction are the findings for $\tau$ (not shown here) and $H$ with the saturation occuring at $\sim 60$ and $\sim 0.75\,\mathrm{km}$, respectively. This is no suprise considering their relation to $Q_L$ (see Eqs. 10 and 15). From Eq. (14), one could expect a similar correlation between the SW cloud radiative effect and the effective radius, but a Spearman correlation below $0.21$ (in absolute values) is found for both the BOA and TOA. This is no suprise considering the derivation of the effective radius by the droplet number concentration (see Eq. 4) and the two modes that are clearly seen in panels (c) and (g) of Fig. 6. The correlations of the SW CRE with the cloud borders and $f_{ad}$ are very weak. In the LW radiation, changes in $Q_L$ (and, thus, in $\tau$ and $H$) possess only a minor influence on CREs (see Table 7) with Spearman (Pearson) correlations below $0.34$ $(0.15)$. In addition, effective radius and droplet number concentration have a moderate effect on the CRE; correlations are below $0.7$. The cloud radiative effect in the LW is mostly dependent on the macrophysical cloud properties, namely the cloud position and vertical extention that impacts the cloud temperature (see Table 7 and Fig. 7) with Spearman (Pearson) correlations above $0.76$ $(0.9)$ in absolute values, following the first rotational component (RC-1, see Table 3). Thereby, we further examined the relation between the first two rotational components and the cloud radiative effects. Confirming our assumption, in Fig. 8, an





excellent monotonic relation is found between $CRE_{SW}$ and RC-2 that is comprised by $\tau$, $Q_L$, and $H$, while a strong linear relation is obtained between $CRE_{LW}$ and RC-1, which is described by CBH and CTH. Even the corresponding densities follow similar patterns, e.g., Fig. 6 panels (a) or (b) with Fig. 8 panel (b) and Fig. 7 panel (b) with Fig. 8 panel (d). The resulting Spearman and Pearson correlations are larger than 0.96 and 0.91, respectively. To this end, such a statistical approach, i.e.,

principal component analysis (plus varimax rotation), can be used as an alternative concept for describing the low-level clouds and, consequently, their radiative impact.

As described in the beginning of Sect. 3.1, the mean distribution of $N_{int}$ is comprised by two clear modes centered around $217\,cm^{-3}$ and $686\,cm^{-3}$ (for 3 June 2016). In addition to this, Fig. 6 further supports this finding and further indicates a strong dependency of the CREs on $N_{int}$. Thus, we further separated the cloud profiles according to the latter two clusters using as,

a mid point, a droplet number concentration of $388\,cm^{-3}$. Subsequently, the correlations between the CREs in the shortwave radiation and all the cloud properties have improved significantly in case of clouds that fell into the right part of the distribution of $N_{int}$. For the LW radiation, a similar increase in correlations, but smaller in magnitude as compared to the SW, is found only for CREs at the TOA. For details with respect to the resulting correlations, the reader is referred to Table C2 in Appendix C. In particular, the largest increase in correlations is found bettwen $CRE_{SW}$ and $r_{int}$, with Spearman and Pearson correlations above

$-0.796$ and $-0.82$, respectively. The latter relationship that is more evident at high values of the droplet number concentration can be explained by the first indirect aerosol effect (e.g., Twomey, 1977; Ackerman et al., 2000; Werner et al., 2014).

## 6    Discussion and conclusions

By analyzing simulations of the high-resolution model ICON-LEM, a sensitivity study has been carried out to investigate the suitability of the vertically homogeneous and the sub-adiabatic cloud models to, firstly, serve as conceptual models for the

evaluation of the representation of low-level clouds in ICON-LEM and similar high-resolution models, and to, secondly, capture the relevant properties which determine the cloud radiative effect. Considering the representation of the cloud microphysical processes in ICON-LEM, we additionally have highlighted the differences in cloud radiative effect resulting from the use of a double- instead of a single-moment cloud microphysics scheme.

ICON-LEM, with its high vertical resolution, ranging from $25\,m$ to $70\,m$ within the boundary layer, and from $70\,m$ to $100\,m$

further up until the upper altitude level of our area of study ($4000\,m$), enables the investigation of the vertical distribution of microphysical properties of low-level clouds. Based on six case days, we find that the behavior of modeled liquid water clouds over Germany more closely resembles the sub-adiabatic than the vertically homogeneous one, in agreement with ground-based observational studies over the same are of interest (Merk et al., 2016). A rather large number of vertical profiles of modeled low-level clouds has been considered in this study and supports the use of the sub-adiabatic model as a conceptual

tool for the evaluation of these profiles in high-resolution models, in agreement with previous studies that supported their use in parameterizations in GCMs (Brenguier et al., 2000). According to the sub-adiabatic model, the key cloud properties which determine the cloud optical thickness and, thus, the SW CRE are the liquid water path, the vertically integrated droplet number concentration (over the cloud geometrical extend, in agreement with Han et al., 1998), the sub-adiabatic fraction,



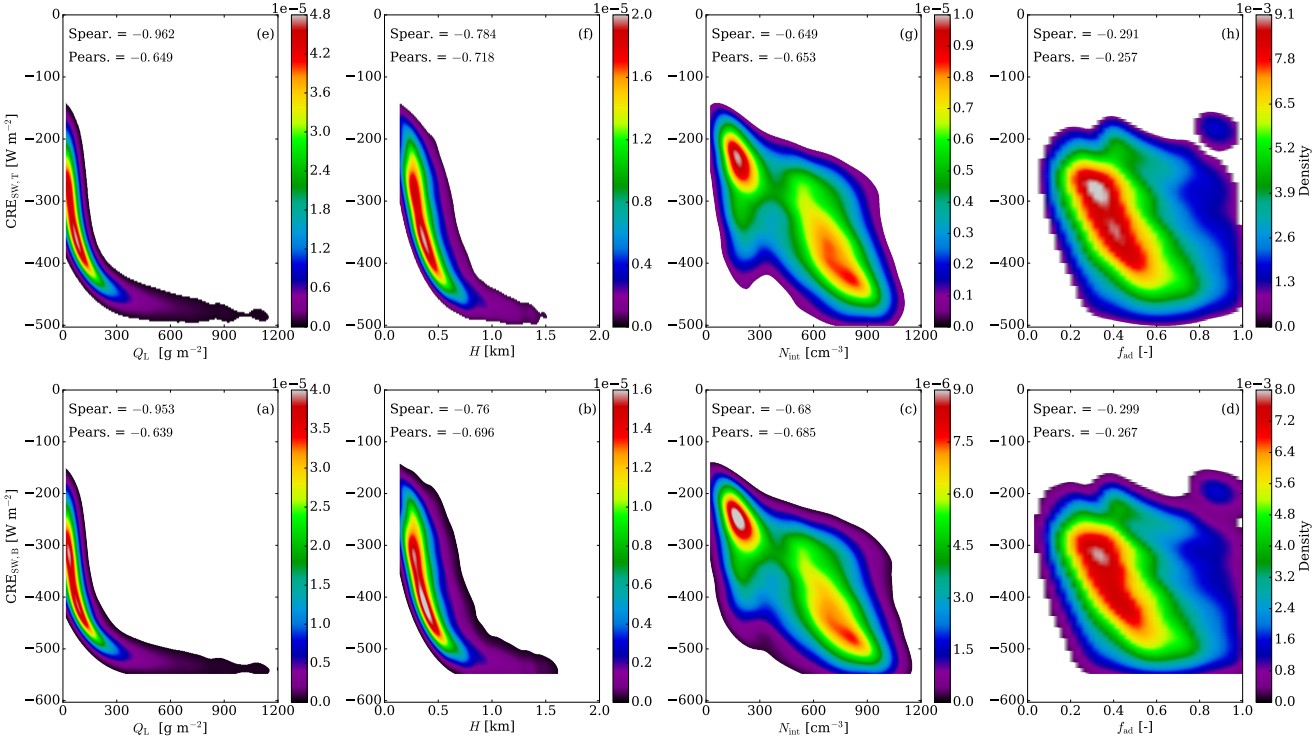

**Figure 6.** Bivariate kernel density (BKD) between the reference simulation (Ref.) and the cloud properties that are essential for the derivation of the cloud optical thickness that is one of the fundamental properties describing the SW cloud radiative effect. Lower panels illustrate the BKD between the $CRE_{SW,B}$ and (a) $Q_L$, (b) $H$, (c) $N_{int}$, and (d) $f_{ad}$, while, the upper panels the BKD between the $CRE_{SW,T}$ and (e) $Q_L$, (f) $H$, (g) $N_{int}$, and (h) $f_{ad}$. The corresponding Spearman (Spear.) and Pearson (Pears.) correlations are highlighted.

and the cloud geometrical extent, which provide a simplified approximation of the vertical structure of clouds. Consistent with this model, we have demonstrated that the cloud optical thickness varies proportionally to $Q_L^{5/6}$ and not linearly with $Q_L$, as predicted by the vertically homogeneous model that further supports both observational and theoretical studies (e.g., Brenguier et al., 2000; Merk et al., 2016). In addition, we show that for our cases, 95.71 % of the variance in cloud optical

5  thickness is explained by the variance in the liquid water path, while the droplet number concentration and the sub-adiabatic fraction contribute only 3.5 % and 0.14 % to the total variance, respectively, outlining the relative importance of the latter properties for describing the SW radiative effect. The sub-adiabatic fraction of clouds is characterized by a large variability ($f_{ad} = 0.45 \pm 0.21$) that strongly varies from day-to-day, but also within the same day, likely driven by entrainment processes. The latter is in agreement with previous studies based on ground-based observations (e.g., Boers et al., 2006; Kim et al., 2008;

10  Merk et al., 2016). Furthermore, we managed to support the outcome of Min et al. (2012); Merk et al. (2016) that the highest values of adiabaticity is linked with optically and geometrically thin clouds. Considering the aforementioned variability of





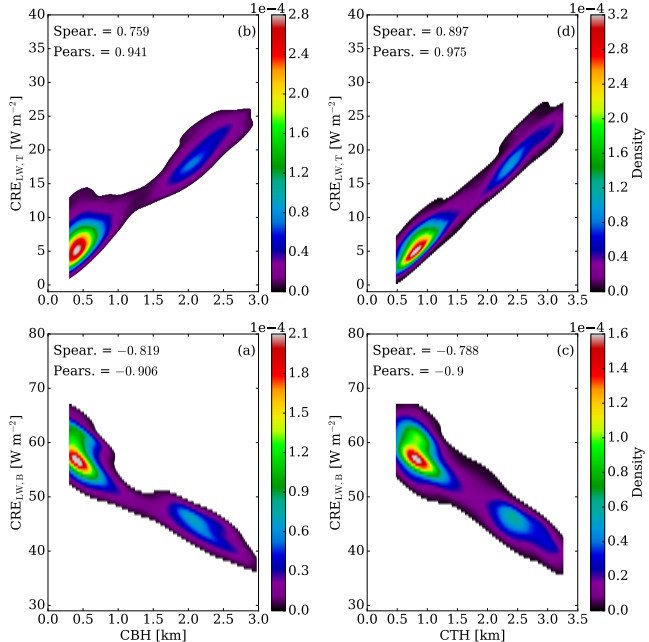

**Figure 7.** Bivariate kernel density (BKD) between the reference simulation (Ref.) and the cloud properties describing the LW cloud radiative effect for the BOA and (a) CBH, (b) CTH, and for the TOA and (c) CBH, (d) CTH. The corresponding Spearman (Spear.) and Pearson (Pears.) correlations are highlighted.

entrainment, the constant and comparatively high values of $f_{\mathrm{ad}}$, which are often adopted in satellite retrievals of cloud droplet number concentration or cloud geometric thickness (e.g., Zeng et al., 2014) are not supported, and might lead to discrepancies in model validation. Therefore, a much lower value of $f_{\mathrm{ad}}$ ranging from 0.4 to 0.6 should be utilized in the sub-adiabatic model to link the cloud optical thickness to the prognostic quantities utilized in GCM parameterizations and determine the indirect

5 effect and cloud feedbacks. The latter value of the sub-adiabatic fraction is close to the one adopted by Grosvenor et al. (2018) for the error assessment of the retrieved $N_{\mathrm{d}}$.

The vertical variability of the droplet number concentration was examined. For 3 June 2016, above an altitude of $2\,\mathrm{km}$, values of $N_{\mathrm{d}}$ lie about $200\,\mathrm{cm}^{-3}$ and are, thus, close to climatological values, while in the boundary layer, the double moment scheme predicts $N_{\mathrm{d}}$ values of about $600\,\mathrm{cm}^{-3}$. Such values are considered rather high compared to satellite remote sensing

10 estimates (Quaas et al., 2006; Grosvenor et al., 2018); in situ observations suggest higher values of $N_{\mathrm{d}}$ and, thus, closer to those simulated by ICON-LEM, but are affected by large instrumental uncertainties (Grosvenor et al., 2018). This identifies a potential weakness in the double-moment scheme and a scrutinization could be useful for further evaluation activities of microphysics parameterization that could lead to better simulations of cloud processes and radiation.





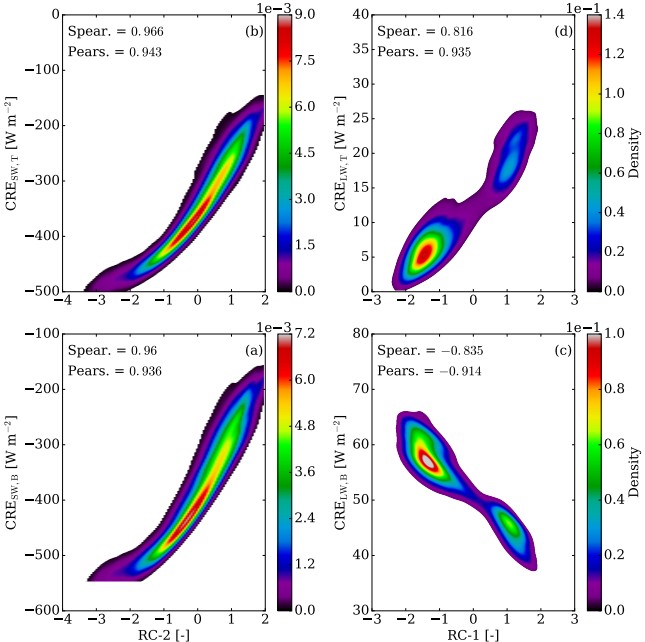

**Figure 8.** For the reference simulation (Ref.), bivariate kernel density (BKD) between $CRE_{SW}$ and the second rotational component (RC-2) at (a) BOA, (b) TOA and between $CRE_{LW}$ and the first rotational component (RC-1) at (c) BOA, (d) TOA. The corresponding Spearman (Spear.) and Pearson (Pears.) correlations are highlighted.

A principal component analysis and a subsequent varimax rotation of cloud properties has been conducted to explore the covariance of cloud properties and radiative effects, and to identify their degrees of freedom and the dominating modes of variability. The goal has been ultimately to uncover potential shortcomings in their representation in models. This analysis reveals that, out of the set of nine parameters considered by us, only four components are sufficient to explain 98 % of the

5 total variance. The first rotational component comprises the cloud bottom and top heights, and thus corresponds to the vertical location of the cloud layer in the atmosphere. The second component combines liquid water path, optical thickness, and geometric extent of the clouds, while the third and fourth component are dominated by the contributions of the sub-adiabatic fraction and the cloud droplet number concentration, respectively. By means of a statistical approach, i.e., principal component analysis (plus varimax rotation), we offer an alternative concept for describing the CREs, with the first and second component

10 representing the main modes of variability determining the LW and SW CREs. While having smaller contributions to the total variance, the third and fourth component are also relevant and potentially capture signatures of the so-called second (cloud geometric extent, Pincus and Baker, 1994) and first indirect aerosol effects (e.g., Twomey, 1977; Ackerman et al., 2000; Werner et al., 2014). This analysis points to the reduced set of parameters for the representation of low-level clouds towards the computation of the CREs: the column effective properties, i.e., $N_{int}$, $Q_L$, $f_{ad}$, $H$, and one of the CTH or CBH. A similar





attempt to provide an alternative concept for the description of the CREs was reported by Schewski and Macke (2003); they tried to correlate domain averaged radiative fluxes from 3D fields with domain averaged properties of cloudy atmospheres.

By means of an offline version of the RRTMG radiative transfer model, idealized simulations have been carried out to estimate the effect of the representation of cloud microphysics in ICON-LEM on the cloud radiative effect; the double-moment

scheme implemented in ICON-LEM (Seifert and Beheng, 2006) has been compared to that of a single-moment scheme. Special emphasis was given on the characterization of the droplet number concentration profile and, thus, the effective radius, that could approximate the microphysical and radiative properties of the modeled low-level clouds as simulated by ICON-LEM (reference scenario). Utilizing a droplet number concentration profile that follows the climatology of coarse atmospheric models (e.g., ECHAM), the single-moment scheme would yield values of the SW CRE which are up to $\sim 39\,\mathrm{W\,m^{-2}}$ less than those of

the double-moment scheme, with a RMSE of $\sim 49\,\mathrm{W\,m^{-2}}$. By employing a more representative profile for the $N_\mathrm{d}$, i.e., two fixed values representing the two modes in the histogram of the droplet number concentration produced by the double-moment scheme leads to a rather good approximation; the RMSE is below $16\,\mathrm{W\,m^{-2}}$. This points to the need to better account for prognostic $N_\mathrm{d}$ calculations.

We investigated the reliability of the vertically homogeneous and the sub-adiabatic model to determine the clouds radiative

effects. The dependency of the differences in CREs (compared to the reference run) on the different droplet number concentration profiles follows the same pattern as for the single- vs double-moment scheme. For the more representative $N_\mathrm{d}$ profiles, the sub-adiabatic cloud model outperforms the vertically homogeneous one for the representation of low-level clouds for calculating their radiative effects and further suggests its use as basis for evaluation of GCM parameterizations, in agreement with (Brenguier et al., 2000).

Based on our results, the following approach is recommended to evaluate the representation of clouds and their radiative effects as simulated by high-resolution atmospheric models: for the shortwave, the vertically integrated water path should be targeted primarily, which is quite reliably retrieved from remote sensing; recent advances in correcting the PP bias enable the retrieval of the liquid water path with high accuracy (Zhang et al., 2016; Werner et al., 2018). In addition, the cloud droplet number concentration and the sub-adiabatic fraction are of relevance and deserve attention, but their reliable derivation remains

challenging both due to the limitations of current remote sensing methods and the lack of validation data on the basis of in situ observations (Grosvenor et al., 2018). In this respect, the rather large values of cloud droplet number concentration reported here as predicted by the two-moment scheme of Seifert and Beheng (2006), $N_\mathrm{d}$ should be scrutinized. For the computation of the cloud radiative effects, the vertical profile of the droplet number concentration is of less importance and fixed profiles could be used, as long as they can represent the different magnitudes in $N_\mathrm{d}$ within and above the boundary layer as shown

here. For the LW CRE, the cloud base and top heights are the determining factors that are rather well derived from ground- and satellite-based observations, respectively. It has be noted, however, that the reliable determination of cloud base height from satellites remains challenging. The sub-adiabatic fraction is also of interest, as it controls the geometric extent of clouds for a given value of liquid water path. Based on our findings, the sub-adiabatic model seems to be better suited than the vertically homogeneous model for the evaluation of the representation of clouds in models.





In future work, the results presented here should be combined with efforts to also take into account the impact of horizontal cloud variability, and in particular of the cloud fraction, which are well-known factors of relevance for the cloud radiative effect. In order to link deficiencies in the CRE to the model representation of cloud properties, an effort should be made to simultaneously evaluate the ICON-LEM-based fluxes and cloud properties discussed here to observations, e.g., through the

combined use of irradiances observed at the top of atmosphere by the Geostationary Earth Radiation Budge (GERB) and at the ground together with measurements of liquid water path, cloud top and bottom height, cloud droplet number concentration, and solar fluxes. This requires the synergistic combination of active and passive remote sensing instruments.

**Appendix A: Derivation of moments of the droplet size distribution**

In Sect. 2.1, the generalized gamma distribution describing the mass of hydrometeors was introduced (see Eq. 1). The $\eta$th

moment is computed by,

$$M_{\mathrm{m}}^{\eta} = A_{\mathrm{m}} \frac{\Gamma(\frac{\eta+\nu+1}{\xi})}{\xi \cdot B_{\mathrm{m}}^{(\frac{\eta+\nu+1}{\xi})}}. \tag{A1}$$

$\Gamma$ stands for the gamma function. For cloud droplets $\nu = \xi = 1$ (see Table 1 in Seifert and Beheng, 2006), the zeroth and first moments of the mass size distribution that denote the droplet number concentration and the liquid water content, respectively, are derived,

$$M_{\mathrm{m}}^{0} = A_{\mathrm{m}} \frac{\Gamma(2)}{B_{\mathrm{m}}^{2}} = N_{\mathrm{d}}, \tag{A2}$$

and

$$M_{\mathrm{m}}^{1} = A_{\mathrm{m}} \frac{\Gamma(3)}{B_{\mathrm{m}}^{3}} = q_{\mathrm{L}}. \tag{A3}$$

Dividing Eq. (A2) by Eq. (A3), one can obtain,

$$B_{\mathrm{m}} = \frac{2 \cdot N_{\mathrm{d}}}{q_{\mathrm{L}}}. \tag{A4}$$

Inserting Eq. (A4) in Eq. (A2) and rearranging gives,

$$A_{\mathrm{m}} = \frac{4 \cdot N_{\mathrm{d}}^{3}}{q_{\mathrm{L}}^{2}}. \tag{A5}$$

According to Seifert and Beheng (2006) and Petty and Huang (2011), a power law is applied for the mass-size relation,

$$x_{\mathrm{m}} = \alpha \cdot \mathrm{d}x = \alpha \cdot b \cdot D^{b-1} \mathrm{d}D. \tag{A6}$$

$D$ denotes the geometrical diameter. In case of spherical particles, $\alpha = \frac{\pi \cdot \rho_{\mathrm{w}}}{6}$ and $b = 3$, with $\rho_{\mathrm{w}}$ being the water density. In

Table 2 in Petty and Huang (2011), one can find the transformation factors between the mass of hydrometers and the diameter





of the hydrometers,

$$A = b \cdot A_{\mathrm{m}} \cdot \alpha^{\nu}, \tag{A7}$$

$$\beta = b(\nu + 1) - 1, \tag{A8}$$

$$B = B_{\mathrm{m}} \cdot \alpha^{\nu}, \tag{A9}$$

5  $$\mu = b \cdot \nu. \tag{A10}$$

Given the aforementioned relations, the formula describing the modified gamma distribution of the DSD is,

$$n(D) = A \cdot D^{\beta} \cdot \exp(-B \cdot D). \tag{A11}$$

Accordingly, the $\eta$th moments of the DSD are given by,

$$M^{\eta} = A \frac{\Gamma(\eta + \beta + 1)}{B^{(\eta + \beta + 1)}}. \tag{A12}$$

10  For the reconstructed DSD, $n(D)$, the zeroth moment ($M^0$) stands for the droplet number concentration. The volume-equivalent radius, $r_{\mathrm{V}}$, is derived from the third moment,

$$r_{\mathrm{V}} = \frac{1}{2} \sqrt[3]{\frac{\int_0^\infty n(D)(D)^3 \, \mathrm{d}D}{(D)^0}}. \tag{A13}$$

**Appendix B: Figures**

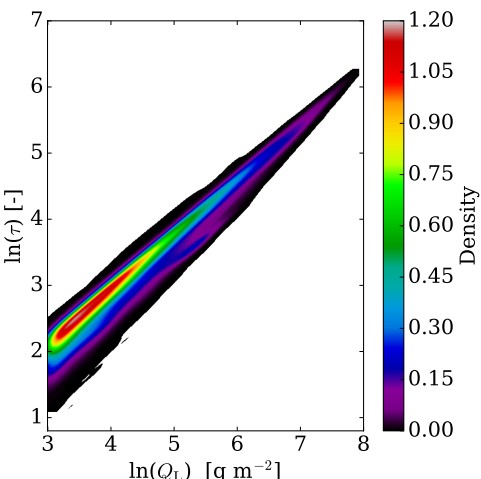

**Figure B1.** Bivariate kernel density (BKD) between the cloud optical thickness and the liquid water path on a logarithmic scale.



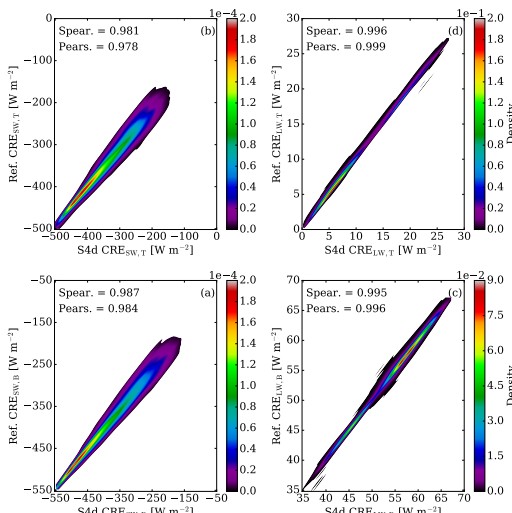

**Figure B2.** Bivariate kernel density (BKD) between the reference simulation (Ref.) and the modified sub-adiabatic run (S4d) in case of the droplet number concentration representing the two clusters in the histogram of $N_{int}$ (see Fig. 1). For the CREs, BKD are presented for the SW radiation at the BOA (a), TOA (b), and for the LW radiation at the BOA (c) and TOA (d). The corresponding Spearman (Spear.) and Pearson (Pears.) correlations are highlighted.

**Appendix C: Tables**





**Table C1.** Mean CRE ($\mathrm{W\,m^{-2}}$) for the LW radiation. Results are given as differences between the new scenario minus the reference simulation ($\Delta$). The root mean square error (RMSE) in $\mathrm{W\,m^{-2}}$ and the Pearson (Pears.) correlation between the new scenarios and the reference simulation are also given.

| Scen. | $\mathrm{CRE_{LW,B}}$ | | | $\mathrm{CRE_{LW,T}}$ | | |
|---|---|---|---|---|---|---|
| | $\Delta$ | RMSE | Pears. | $\Delta$ | RMSE | Pears. |
| S2 | $-0.10$ | 0.32 | 0.999 | $-0.02$ | 0.07 | 1.000 |
| S2b | $-0.06$ | 0.22 | 0.999 | $-0.03$ | 0.08 | 1.000 |
| S2c | $-0.02$ | 0.30 | 0.999 | 0.00 | 0.09 | 1.000 |
| S2d | $-0.04$ | 0.28 | 0.999 | $-0.02$ | 0.08 | 1.000 |
| S3a | 0.40 | 0.71 | 0.993 | 0.20 | 0.38 | 0.997 |
| S3b | 0.48 | 0.74 | 0.994 | 0.22 | 0.38 | 0.997 |
| S3c | 0.51 | 0.76 | 0.994 | 0.24 | 0.39 | 0.997 |
| S3d | 0.50 | 0.75 | 0.994 | 0.23 | 0.39 | 0.997 |
| S4a | $-0.03$ | 0.57 | 0.995 | 0.29 | 0.47 | 0.996 |
| S4b | $-0.01$ | 0.59 | 0.995 | 0.30 | 0.48 | 0.996 |
| S4c | 0.07 | 0.57 | 0.995 | 0.32 | 0.48 | 0.996 |
| S4d | $-0.01$ | 0.60 | 0.994 | 0.31 | 0.48 | 0.996 |
| S5a | 0.12 | 0.58 | 0.994 | 0.27 | 0.44 | 0.996 |
| S5b | 0.19 | 0.58 | 0.995 | 0.28 | 0.44 | 0.996 |
| S5c | 0.23 | 0.58 | 0.995 | 0.30 | 0.45 | 0.996 |
| S5d | 0.19 | 0.59 | 0.995 | 0.29 | 0.45 | 0.996 |

**Table C2.** Spearman and Pearson (Spearman/Pearson) between the cloud radiative effects and the cloud properties for the two major clusters characterized by low $N_{\mathrm{int}}$ values (L) and high $N_{\mathrm{int}}$ values (H).

| Properties | $\mathrm{CRE_{SW,B}}$ | | $\mathrm{CRE_{SW,T}}$ | | $\mathrm{CRE_{LW,B}}$ | | $\mathrm{CRE_{LW,T}}$ | |
|---|---|---|---|---|---|---|---|---|
| | L | H | L | H | L | H | L | H |
| $Q_{\mathrm{L}}$ | $-0.919/-0.784$ | $-0.991/-0.672$ | $-0.923/-0.784$ | $-0.988/-0.682$ | $0.181/0.011$ | $-0.112/-0.284$ | $0.243/0.179$ | $0.620/0.682$ |
| $\tau$ | $-0.994/-0.841$ | $-0.999/-0.729$ | $-0.988/-0.836$ | $-0.997/-0.738$ | $0.305/0.110$ | $-0.133/-0.282$ | $0.0640.076$ | $0.613/0.685$ |
| $N_{\mathrm{int}}$ | $-0.408/-0.380$ | $-0.464/-0.463$ | $-0.352/-0.318$ | $-0.467/-0.465$ | $0.654/0.663$ | $-0.134/-0.105$ | $-0.670/-0.680$ | $0.161/0.113$ |
| $r_{\mathrm{int}}$ | $-0.251/-0.325$ | $-0.830-0.802$ | $-0.278/-0.352$ | $-0.820/-0.796$ | $-0.319/-0.347$ | $-0.042/-0.106$ | $0.614/0.567$ | $0.572/0.647$ |
| CBH | $0.223/0.165$ | $-0.580/-0.097$ | $0.133/0.062$ | $0.085/-0.132$ | $-0.872/-0.895$ | $-0.612/-0.654$ | $0.907/0.958$ | $0.397/0.601$ |
| CTH | $0.015/0.027$ | $-0.731/-0.635$ | $-0.076/-0.078$ | $-0.751/-0.665$ | $-0.837/-0.888$ | $-0.517/-0.600$ | $0.949/0.975$ | $0.756/0.874$ |
| $H$ | $-0.688/-0.703$ | $-0.900/-0.772$ | $-0.713/-0.724$ | $-0.904/-0.784$ | $-0.025/-0.065$ | $-0.192/-0.287$ | $0.270/0.196$ | $0.585/0.696$ |
| $f_{\mathrm{ad}}$ | $-0.042/-0.042$ | $-0.366/-0.339$ | $-0.038/-0.034$ | $-0.354/-0.325$ | $-0.042/-0.041$ | $0.082/0.071$ | $0.159/0.148$ | $0.226/0.215$ |



*Data availability.* The full 3D large eddy simulation fields used for this paper are stored at the Deutsche Klima Rechenzentrum archive (DKRZ) as part of the HD(CP)2 project.

*Code and data availability.* The Python RRTMG interface (pyRRTMG) is available online at https://github.com/hdeneke/pyRRTMG.

*Author contributions.* VB conceived and refined the overall structure of the investigation, based on discussions with and feedback from all
5   co-authors. VB carried out and refined the data analysis. HD implemented the Python interface to RRTMG used in the analysis. VB wrote the draft manuscript, with all authors contributing to the interpretation of the results and to its improvement.

*Competing interests.* The authors declare that they have no conflict of interest.

*Acknowledgements.* This work has been conducted in the framework of the High Definition Clouds and Precipitation for Advancing Climate Prediction HD(CP)$^2$, funded by the German Federal Ministry of Education and Research BMBF under grant no. 01LK1504B. We thank our
10   colleagues, Anja Hünerbein and Frank Werner, for the many thoughtful comments that led to the improvement of the manuscript.



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
