# Peer review of "The sub-adiabatic model as a concept for evaluating the representation and radiative effects of low-level clouds in a high-resolution atmospheric model"

_Atmospheric Chemistry and Physics, 2019_

## Referee Comment (RC1) · Anonymous Referee #3 · 18 Jul 2019

MS No.: acp-2019-137
Title: The sub-adiabatic model as a concept for evaluating the representation and radiative effects of low-level clouds in a high-resolution atmospheric model
Authors: Barlakas, V., H. Deneke, and A. Macke

[Figure]

**General comments**

This paper aims to propose a procedure for evaluating the radiative effect of low-level clouds using high-resolution simulations. The authors try to clarify the relationship between several cloud properties and the cloud radiative effect. This kind of study is considered to be important to understand the uncertain role of clouds in the atmospheric climate system.

However, the robustness of the obtained results, such as principal components of cloud properties and their relationship with the CRE, should be discussed. These results are obtained only from the one-day data. However, the daily variation is large as shown in Table. 1, although the authors claim that the day has similar properties to the six-day average.

In addition, there are many mistakes such as the values between in the text and tables do not match. Careful preparation of the manuscript is required before submitting the manuscript.

**Specific comments**

- Order of diagrams in Figures

    The order of diagrams (a,b,c,...) is not uniform for the figures.
    The authors must use the identical order to avoid reader's confusion.

- Significant figures of values

    Several values seem to have too many significant figures, such as 0.9921

- Section 2.1

Brief descriptions of the model and configurations of the experiment are necessary, such as kind of governing equations, vertical levels, and calculation domains.

- Equation 15

  The fact 9/5 seems to be 3/2.
  The power of $(18\rho_w^4 f_{ad}\Gamma_{ad})$ is not 1/6 but -1/6.

- Page 9, Line 18, A close relation between the effective radius and the droplet number concentration exist.

  Why does the effective radius have a single-mode distribution in spite of the bimodal distribution of the droplet number concentration?
  "exist" -> exists

- Figure 2 and Figure 3 (b)

  Both diagrams show the distribution of the Nd, the magnitude of the median is quite different. What makes such a big difference?

- Section 3.3.1

  There is a large relationship between $Q_L$ and $f_{ad}$, and then this analysis has multicollinearity problem. Therefore, the amount obtained must be much interpolated more carefully. Furthermore, I do not agree that the $\tau$ is proportional to $Q_L^{5/6}$, since Eq. (15) has $f_{ad}$.

- Page 17 Line 19, resulting in a net cooling.

  There exists large uncertainty, and I wondered if the negative value has statistical significance.

- Page 18 Line 1, Table 6 lists the mean CREs between . . .
It should be "Table 6 lists the difference of the mean CREs between . . .".
The same corrections are necessary for the following sentences.

- Page 18 Line 8, For a given liquid water path, the smaller . . .

  Check if it is grammatically correct.

- Page 18 Line 16, a Pearson correlation of 0.950 (0.928) is yielded.

  The values are 0.952 (0.930) in Table 6.

- Page 18 Line 19, a Pearson correlation of 0.995

  The value is 0.996 in Table 6.

- Page 18 Line 19, and Page 20 Line 13, no suprise considering

  surprise

- Page 18 Line 25, of about -6.52 Wm$^{-2}$ with a RMSE of 10.4 Wm$^{-2}$ for b and -9.31 Wm$^{-2}$ with a RMSE of 19.4Wm$^{-2}$ for d

  The sign of -6.52 and -9.31 is different from that in Table 6.
  The mismatch is also in the number of -0.11 and -3.64 at Page 19 Line 1.

- Table 6.

  The names of Scen. are wrong.

- Page 19 Line 3, For instance, in case of the adiabatic scenarios . . .

  the sub-adiabatic

- Page 19 Line 7, slightly larger scatter is found for S4 as compared to S3.

  Why is the result of S4 worth than that of S3?

- Page 20 Line 7, -0.76

  It has different significant figures from that in Table 7.
  Same for 0.21 at Page 20 Line 13

- Page 20 Line 9, The latter monotonic relation that is found stronger for lower values of the liquid water path saturates at $Q_L > 300\,\mathrm{gm}^{-2}$.

  Logarithmic axis is preferred. The saturation may not be found in the logarithmic plot.

- Page 21 Line 3, e.g., Fig. 6 panels (a) or (b) with Fig. 8 panel (b)

  Fig. 8 panel (a)

- Page 21 Line 3, The resulting Spearman and Pearson correlations larger than 0.96 and 0.91, respectively.

  The values seem to be 0.816 and 0.914.

- Page 21 Line 14, with Spearman and Pearson correlations above -0.796 and -0.82, respectively.

  The values should be -0.820 and -0.796.
  The author should mention that these values are only for high values of the droplet number concentration.

---

## Referee Comment (RC2) · Anonymous Referee #1 · 24 Jul 2019

General comments:

The authors investigated a sensitivity of the vertical distribution of the assumed adiabaticity (i.e. sub-adiabatic versus vertical homogeneous cloud model) to their cloud radiative effect of low-level clouds. The effect of the different adiabatic model applied was examined by using ICON-LEM, and also evaluated differences in the cloud properties by switching from single-moment to double-moment microphysics scheme in their model. They conducted simulations on six case days and found that the sub-adiabatic model resembles better characteristics of liquid clouds rather than the vertically homo-

geneous assumption.

The authors have revised several minor issues according to the referee comments in access review before ACPD. While the manuscript is scientifically sounds and is publishable with further improvements and clarification, I still feel that the manuscript does not reach the standard of ACP.

My major concern is that the simulations examined in this study are very limited (Page 8 Line 15-18). The authors conducted simulations using six case days, but actually looked at in details only the case of 3 June 2016. How general are they? Doesn't the vertical structure of adiabaticity depend strongly on the cloud regimes and types or their life-stage? In the present form of this paper, objectives are too narrow. The described relationship among cloud micro- and macrophysical properties and radiative effect using high resolution simulation may provide key suggestions on aerosol-cloud interactions, but the findings as they are, are by no means general. With some more simulation cases or a bit more analysis for all the case days in detail, I think this will make a publishable work.

Specific comments:

Section 2.3: Please describe the model resolution, domain size, as well as timestep used in the simulations. The general description of ICON-LEM on page 3 (lines 16-17 and 28-30) is confusing with regards to this.

Equation (7): It is better to add a sentence about the factor 2/3, rather than 5/9, citing relevant papers (e.g., Szczodrak et al., 2001; Wood and Hartmann, 2006; Lebsock and Su, 2014). Equations (14) and (15) as well.

Figure 3 and caption: qL -> QL or CLWP

Figures 6, 7 and 8: The order of subfigures is not consistent with the caption.

Table 6: I found several mismatches between Table 6 and citing main text (e.g., page 18 line 16), which made reviewers very difficult to track...

Page 23 Line 11-13: This sentence is too vague. Please raise more specific source of uncertainty, and describe how the scrutinization is required.

Page 25 Line 12-13: This sentence recommends double-moment cloud microphysics, but page 23 line 12 points weakness of the double-moment.

Appendix B: Please change the appendix title. Appendix section is not just a list of supporting materials. The current version does not have any explanation about the figures in the appendix (Appendix C as well).

---

## Referee Comment (RC3) · Anonymous Referee #2 · 7 Aug 2019

Summary Barlakas et al. exhaustively analyze simulated shallow cloud profiles over Germany from a few select days to analyze cloud radiative effects (CREs) and the minimal set of parameters necessary to represent these effects. A number of different complementary and at times overlapping analyses are performed to isolate the optimal CRE parameter set, which are generally in close agreement with past observational and modeling studies.

Due to major concerns regarding the novel contributions of this study, I recommend major revisions before publication. The authors may consider trimming the manuscript

to be considerably shorter given the redundancy of many of their results.

General comments I have serious concerns about the novel contributions of this study and I found that the results of the work undermined the stated motivation: 1. There is already a significant body of work on the topic of sub-adiabaticity (much of which the authors cite) and the results of this study seem to confirm past findings (in particular, those of Merk et al., 2016) with little added insight into process (radiative, microphysical, etc.) besides pointing out that single-moment microphysics schemes leave much to be desired (which has been explored by e.g., Igel et al., 2015, JAS). Much work has also been done with respect to statistical emulators for understanding cloud radiative effects (e.g. Feingold et al., 2016; Glassmeier et al., 2019) and the aggregation of model data over all shallow cloudy columns severely limited the authors' ability to examine details regarding differences between cumulus and stratus (which likely exhibit very different fad), the diurnal cycle, or radiative effects across spatial scales – an exploration of the latter would be especially useful since the 300 m HOPE dataset is finer in horizontal resolution than the existing remote sensing products this study is designed to improve (typically ∼1 km pixel size).

Finally, I get the sense that this paper only deals with sub-adiabaticity in passing – the latter half of the paper is primarily concerned with describing CREs with a minimal set of variables and is almost completely disconnected from the title of the paper. Sub-adiabaticity seems to have only a weak influence on CREs.

2. With respect to motivation, the authors rely heavily and repeatedly on the idea that there are large uncertainties in aircraft measurements of cloud drop number concentration (ND in the authors' notation), which they justify by citing the ND retrieval review paper of Grosvenor et al. (2018) – specifically, I believe they refer to Grosvenor et al.'s Figure 5 (which is in turn based on data used in Siebert et al., 2013) and accompanying discussion. This is an unfortunate figure. The disagreement of two probes (Phase Doppler Interferometer and Particulate Volume Monitor; PDI and PVM, respectively) at concentrations of ND >350 cm-3 is used as evidence that in situ probes have a general,

systematic problem measuring ND.

The issue with this illustration is that one of the two probes used (PVM) is not designed to measure ND and I am aware of no other publication in which this is even attempted. The PVM measures extinction from a population of cloud drops and makes no explicit count of particle density. In fact, I'm not even sure how this quantity was generated since the PVM returns only two data streams: total particle volume and surface area. The PDI, on the other hand, is frequently used by both the airborne cloud physics and industrial spray characterization communities and has been demonstrated to accurately count (and size) particles up to a concentration of $O(10^5)$ cm-3. An intercomparison of PDI with other probes that explicitly count particles (CAS, FSSP, CDP, Holodec. . .etc. – there are a great number and I don't understand why Grosvenor et al. chose such an ill-suited probe for their figure) would likely show a much better overlap in the PDFs of ND from different probes; such an intercomparison of the latest generation of cloud probes is currently underway for the recent NASA ORACLES campaign, which sampled a wide variety of concentration conditions due to the campaign's focus on interaction of clouds with overlying smoke layers during the stratocumulus to cumulus transition.

I am strongly opposed to the use of phrasing such as "large instrumental uncertainties" (e.g. page 23, lines 10-11) as I think this point is vastly overstated by Grosvenor et al. (2018), an assertion backed by their discussion of myriad other issues with retrieval assumptions ahead of any problems with in situ measurements.

In the remainder of the review, specific comments reference "PX, LY" where X is page number and Y is line number. When a direct edit to the text is suggested, it is given in italics. (see attached document)

Specific comments

P2, L7: "taking placed" should be "taking place"

P2, L21-22: "fixed droplet number distribution" – ambiguous terminology; "fixed droplet size distribution" would be clearer.

P2, L23: "Double-moment microphysical schemes. . .are only recently becoming more widespread": Perhaps in the operational forecasting community this is true, but in research modeling (especially of warm clouds), double-moment schemes have been common for at least a decade.

P5, L16: Why do you use an indirect measure for rain/drizzle instead of directly examining rain water mixing ratio? I understand that it makes for a more straightforward comparison with observations, but it seems like an unnecessary step.

P6, L2: "The model outputs the. . ."

P6, L19: Is the assumption of vertical homogeneity a "scheme?" Seems like an odd word choice.

P7, L15: "Clapeyron relationship"

P7, L16-17: This sentence is difficult to follow. Rephrase and simplify the structure for clarity.

P11, L8: Remove "the" from "the 5 May. . ."

P13, L12-13: "with a 5/6 slope" – possibly remove the word "fit," doesn't make sense in context

P13, L18: If fad only accounts for 0.14% of the variance in ïĄť, what's the point of all this?

P13, Section 4: The step by step narrative of the PC analysis is overwrought. If you primarily intend to use the results of the RC analysis to justify the minimal set of variables needed to represent CREs, skip the PC discussion; the PC and RC results are sufficiently similar that it is redundant. P13, L34: "optimal" instead of "optimized"

[Figure]

P14, Table 3 caption: Remove trailing zero from "moderate [0.40, 0.6)" for consistency

P14, L7-9: Rearrange sentence beginning "However, the PCs..." to simplify structure for clarity.

P14, L11: remove "so-called" – this makes it look like other people have a different name for it

P15, L9-10: I am confused by what you're doing here – are you always running multiple simulations, or for scenarios S2-S4 are you imposing LWC/ND profiles that are not actually from the simulations?

P15-16, L33-3: the assumptions would be more clearly expressed in a table

P16, L1-2: Why are there drops in the free troposphere?

P16, L3: "where the liquid water path is preserved"

P16, L8: "following the climatology of a coarse..." – you only use the ECHAM value. Is this representative of what all GCMs do? If not, the generalization doesn't work.

P17, Section 5.1.2: I found the latter half of this discussion to be very difficult to follow, especially the references to various scenarios by only a letter or number near the end of the section (i.e. last paragraph, P18).

P18, Table 5 caption: Cosine SZA was just given in text (and will hopefully be put in a separate table of assumptions) – remove since redundant.

P18, L1: "and the rest of the simulated..."

P19, Table 6: Two things: 1) numbering of scenarios is off by one and 2) since BOA and TOA are almost always within 5% or 1 W/m2 of each other, can you just pick one and reduce the amount of information here? This table would be much more effective/digestible.

P19, L11-14: You can test whether effective radius is outside the range. Is this an issue

or isn't it?

P20, Section 5.1.3: As with the PCA results, what is the point of showing both correlations? You almost exclusively discuss Spearman, so why not just show that?

P20, L7: Capitalize "Spearman"

P24, L3: "uncover potential shortcomings in…models": you only compared the model to itself, so how did you uncover shortcomings? Do you mean LES vs. GCM? Beyond discussing single- vs. double-moment microphysics (an already well-known issue), what shortcomings did you uncover?

P24, L11: quantify contributions of 3rd/4th components to total variance here.

P24, L11: delete "so-called"

P25, L9: again, is the ECHAM climatological ND representative? Is it even backed by observations? You have not made a case for why this is a good number to use, besides the fact that a single GCM uses it.

P25, L10: How do two fixed values constitute a profile?

P27, Eq A13: is exponent in denominator a typo? $D\hat{\ }0=1$.

Please also note the supplement to this comment:
https://www.atmos-chem-phys-discuss.net/acp-2019-137/acp-2019-137-RC3-supplement.pdf

———————————————

---

## Author Comment (AC1) · 8 Oct 2019

**Answers to Anonymous Referee #3**

We thank the anonymous referee #3 for his/her constructive comments and suggestions that certainly have improved the manuscript significantly. We revised the manuscript according to his/her comments and the comments of anonymous referees #1 and #2. In the following,

- *referee's comments are given in italic,*

- our answers are outlined in normal format, and

- **textual changes in the manuscript are given in bold format**.

We would like inform the anonymous referee #3 about the following changes:

1. Driven by the specific comment (SC) #18 of anonymous referee #3 (SC3.18), we decided to drop scenario S4 from the analysis. The difference between the sub-adiabatic model (S3) and the modified one (S4) is that the latter accounts for the depletion of the liquid water content due to entrainment, precipitation, and freezing drops. Consequently, we wanted to check whether S4 captures better the vertical stratification of the modeled low-level clouds and, accordingly, if it approximates the CREs of the reference simulation with better accuracy. Since S4 does not provide any further insight, we now have decided to drop this scenario. However, we do confirm that, by considering all the case days in the analysis, we came to the same conclusions as for 3 June. As a confirmation, we updated the Tables and attached them at the end of this document. The referee is referred to Tables R1–R3.

2. In all scenarios, we decided to drop sub-case d, which employs two fixed values for the droplet number concentration representing the two modes in the corresponding histogram for 3 June 2016. This scenario separates clouds into a cluster with low/high clouds. Considering the vertical variability of the droplet number concentration, the latter clustering will link low clouds (within the boundary layer) with high $N_{\mathrm{d}}$ and, accordingly, high clouds with lower $N_{\mathrm{d}}$ values. Thus, for all scenarios, employing such values for $N_{\mathrm{d}}$ are able to approximate the reference radiative transfer simulation very well. Only the radiative transfer simulation that is supplied by the droplet number concentration weighted over the cloud geometrical extent, i.e., $N_{\mathrm{int}}$ (sub-case b) leads to smaller differences when compared to the reference simulation. However, we do confirm that, by considering all the case days into the analysis, we came to the same conclusions as for 3 June. Note that, for the latter case, the clustering was conducted on the mean $N_{\mathrm{int}}$ over all case days. As a confirmation, we updated the Tables and attached them at the end of this document. The referee is referred to Tables R1–R3.

3. We decided to add a new scenario as a replacement of sub-case d, whereby radiative transfer simulations are conducted for a mean vertical profile of the droplet number concentration over all case days. Tables R9–R11 summarize the new results. In brief, this scenario is considered as an improvement compared to the clustering case. The following parts were included within the text:

   Section 5.1.2: **Last but not least, by replacing the vertical profile of $N_{\mathrm{d}}$ by the**

mean profile of $N_\mathrm{d}$ over all case days (see Fig. 2), emulates the cloud radiative effects of the reference simulation quite well. Accordinly, scenario S4 slightly undersimates the mean SW CREs, with an mean error up to $-3.16\,\mathrm{W\,m^{-2}}$ and a RMSE up to $17.2\,\mathrm{W\,m^{-2}}$ for both BOA and TOA. In fact, this scenario outperforms the rest scenarios (S1–S3), except from the sub-case b ($N_\mathrm{int}$) in all scenarios. For an illustration of the excellent linear correlation between the reference simulation and S4 by means of a bivariate kernel density (BKD) plot, the reader is referred to Fig. B1 in Appendix B. One can see that the CREs computed by these scenarios are in a very good agreement almost everywhere except towards larger values of the CREs in case of the SW radiation, with Pearson correlations larger than 0.977 for both BOA and TOA.

Section 6: **By employing a more representative profile for the $N_\mathrm{d}$, i.e., a mean vertical profile of $N_\mathrm{d}$ over all case days leads to a rather good approximation; the RMSE is below $17.2\,\mathrm{W\,m^{-2}}$. This points to the need to better account for prognostic $N_\mathrm{d}$ calculations.**

Appendix B: **In sect. 5.1.2, by conducting idealized radiative transfer simulations, we estimated the impact of the representation of cloud properties in ICON-LEM on the cloud radiative effects (CREs). Special emphasis was given on identifying the droplet number concentration ($N_\mathrm{d}$), which approximates the microphysical and radiative properties of low-level clouds as simulated by ICON-LEM (reference scenario). A radiative transfer simulation, which employs a mean vertical profile of $N_\mathrm{d}$ over all the case days (scenario S4), approximates the CREs of the reference scenario quite well. Figure B1 depicts the excellent linear correlation between the reference simulation and S4 by means of a bivariate kernel density (BKD).**

4. Following the general comment of anonymous referee #2 for shortening the manuscript given the redundancy of many of the results shown in this study and his/her relevant specific comments (SC), i.e., (SC2.12) and (SC2.25):

   • We decided to drop Fig. B1. Figure B1 illustrates the bivariate kernel density (BKD) between the cloud optical thickness and the liquid water path on a logarithmic scale. Considering the comprehensive explanation given in Sect. 3.3.1, we decided that this illustration did not provide any additional information.

   • Figures 6 and 7 have been revised. Now, they illustrate results only for TOA (see Figs R2 and R3).

   • We now focus only on the rotational component analysis. The mention of the principal component analysis have been significantly reduced. In addition, we removed the relevant information from Table 3. For the updated version of the Table, the referees are referred to Table R5. Additionally, we replaced Figure 5 by Table R4. This table lists the contribution of each rotational component to the total variance.

**Answers to general comments (GC) from referee #3 (GC3)**

(**GC3.1**) *The robustness of the obtained results, such as principal components of cloud properties and their relationship with the CRE, should be discussed. These results are obtained only from the one-day data. However, the daily variation is large as shown in Table. 1, although the authors claim that the day has similar properties to the six-day average.*

We revised our manuscript according to the comments of anonymous referee #3 and the comments of anonymous referees #1 and #2. We further extended our analysis over all case days to improve the robustness of our results. Now, sections 3.2 and 5 outline our findings for all case days.

**Answers to specific comments (SC) from referee #3 (SC3)**

(**SC3.1**) *Order of diagrams in Figures*

The order of sub-figures has been revised.

(**SC3.2**) *Significant figures of values.*

The number of significant figures of values has been revised.

(**SC3.3**) *Brief descriptions of the model and configurations of the experiment are necessary, such as kind of governing equations, vertical levels, and calculation domains.*

We revised the description of ICON-LEM (Section 2.1) according to the anonymous referee 3 and the specific comment (SC3) of anonymous referee 1:.

The ICON unified modeling framework was co-developed by the German meteorological service (DWD) and the Max Planck institute for meteorology (MPI-M) in order to support climate research and weather forecasting. Within the HD(CP)2 project, ICON was further extended towards large eddy simulations with realistic topography **and** open boundary conditions. This resulted in ICON-LEM **deployed in restricted areas that are centered on Germany and the Tropical Atlantic [1]. The equations utilized by the model are based on the prognostic variables given by Gassmann and Herzog [2]. These variables comprise the horizontal and vertical velocity components, the density of moist air, the virtual potential temperature, and the mass and number densities of traces, e.g., specific humidity, liquid water, and different ice hydrometeors. A comprehensive description of the model and its governing equations is found in Dipankar et al. [3] and Wan et al., [4].** Concerning turbulence parameterization, the three-dimensional Smagorinsky scheme is employed [3]. The activation of cloud condensation nuclei (CCN) is based on the parameterization of Seifert and Beheng [5] and modified in order to account for the consumption of CCNs due to their activation into cloud droplets. The CCN concentration is then parameterized following the pressure profile and the vertical velocity [6].

**Simulations are carried out for three different domains with 624 m, 312 m, and 156 m**

horizontal resolution. The model domains consist of 150 vertical levels, with resolutions ranging from ∼25 m to 70 m within the boundary layer, and from 70 m to 355 m further up until the top of the domain at 21 km. For each of the aforementioned grids, **data** is stored as one-dimensional (1D) profiles every 10 sec, two-(2D), and 3D snapshots [1]. The model yields output on each of the aforementioned grids with the data stored as one-dimensional (1D) profiles, two-(2D), and 3D snapshots [1]. In case of the 3D output, the simulation data is interpolated from the original grids (e.g., 156 m) to a 1 km grid, the 3D coarse data, and 300 m grid, the so-called HOPE data. The latter output has been created for the purpose of model evaluation with ground-based observations from the HD(CP)$^2$ Observational Prototype Experiment (HOPE) that took place near Jülich [7]**and is limited to a domain size of about ∼45 km$^2$. Note here that for the 2D and 3D output, data is stored at day- and night-time frequency. Day-time frequency begins at 06:00 UTC and lasts until 00:00 UTC, while night-time starts at midnight and lasts until 06:00 UTC. The 2D data is stored with a day-time and night-time frequency of 10 sec and 5 min, respectively. The 3D coarse data have day-time frequency of 10 min (1 hour at night-time). In this study, the 3D HOPE data has been used that is stored only at a day-time frequency of 15 min.**

**(SC3.4)** *Equation 15. The fact 9/5 seems to be 3/2. The power of (18w4 fad ad) is not 1/6 but -1/6.*

The referee is correct. This section has been revised following also the specific comment #2 of the anonymous referee #1 (SC1.2):

P6 L24: **while the factor $2/3$ is a scale factor resulting from the constant liquid water content and effective radius with height [8].**

P7 L21: **Compared to Eq. (7), Eq. (10) leads to a factor of $5/9$, meaning that the sub-adiabatic liquid water path is $5/6$ times the one of the vertically homogeneous model [9].**

P8 L11: **For vertically constant $q_{\mathrm{L}}$ and $r_{\mathrm{eff}}$, this can be interpreted as the cloud optical thickness coming from the vertical homogeneous model (see Eq. 7). According to the sub-adiabatic cloud model, the cloud optical thickness is linked to the liquid water path and the effective radius [10],**

$$\tau = \frac{9}{5}\frac{Q_{\mathrm{L}}}{\rho_{\mathrm{w}} \cdot r_{\mathrm{eff}}}$$

**Alternatively, substituting $r_{\mathrm{eff}}$ from Eq. (13) in Eq. (15), the cloud optical thickness is given by,...**

**(SC3.5)** *P9 L18: A close relation between the effective radius and the droplet number concentration exist. Why does the effective radius have a single-mode distribution in spite of the bimodal distribution of the droplet number concentration?*

The effective radius is defined as the ratio of the third to the second moments of the droplet size distribution and the second moment of the size distribution is closely related to the liquid water content. Accordingly, the two modes of $N_{\mathrm{d}}$ do not need to be at two different size regimes in $r_{\mathrm{eff}}$.

(**SC3.6**) *P9 L18: "exist" -¿ exists.*

Corrected.

(**SC3.7**) *Figure 2 and Figure 3 (b) Both diagrams show the distribution of the Nd, the magnitude of the median is quite different. What makes such a big difference?*

Figure 2 shows the histogram as a box-whisker plot of the droplet number concentration for each model level. On the other hand, Fig. 3 depicts the mean profile of $N_\mathrm{d}$ normalized over the cloud geometrical extent, illustrating the vertical change in $N_\mathrm{d}$ within individual clouds. The aforementioned normalization is the reason of the differences between the two figures. As we aforementioned, we further extended the analysis over all case days. Thus, fig. (2) and fig. (3) have been revised (see Fig. R1 and Fig. R2).

(**SC3.8**) *Section 3.3.1: There is a large relationship between $Q_\mathrm{L}$ and fad, and then this analysis has multicollinearity problem. Therefore, the amount obtained must be much interpolated more carefully. Furthermore, I do not agree that the  is proportional to $Q_\mathrm{L}^{5/6}$ , since Eq. (15) has $f_\mathrm{ad}$.*

We acknowledge referee's concerns with respect to our multicollinearity analysis. However, we respectfully disagree on this point. In this section, we tried to predict the cloud optical thickness derived from the output of ICON-LEM (by using Eq. 14), via employing the relevant equation suggested by the sub-adiabatic model, i.e., Eq. (15). In the analysis, results from all case days has been considered. By employing the sub-adiabatic model, i.e., model $Y_4(Q_\mathrm{L}, f_\mathrm{ad}, N_\mathrm{int})$, we managed to approximate the cloud optical thickness quite well. In fact, $Y_4$ explains 99.9% of the variance in cloud optical thickness with a root mean square error of 0.027. In addition, we found only a weak correlation between the liquid water path and the sub-adiabatic factor (Pearson correlation of 0.28), in contrast to the very strong correlation between the cloud optical thickness and the liquid water path (Pearson correlation of 0.99). The referee is referred to Figure 4 in section 4.

However, we did revise section 3.1.1, in order to avoid any confusion. The following parts have been included:

P13 L2: **With this intention, an effort has been conducted to predict the cloud optical thickness derived from Eq. (14) by employing the sub-adiabatic model and Eq. (15).**

Caption of Table 2: Prediction of cloud optical thickness by ordinary least squares regression method:

P13 L20: **In fact, model $Y_4(Q_\mathrm{L}, f_\mathrm{ad}, N_\mathrm{int})$ supports the applicability of the sub-adiabatic model since it is able to approximate the cloud optical thickness with high accuracy (RMSE = 0.027)**

(**SC3.9**) *P17 L19: resulting in a net cooling. There exists large uncertainty, and I wondered if the negative value has statistical significance.*

We acknowledge the referee's concerns with regard to the uncertainty in the resulting cloud radiative effects. However, low-level clouds tend to be rather warm and, hence, having a generally small influence on the TOA lowngwave radiation. In contrast, they are characterized by a large albedo, leading to an overall net cooling effect. The latter net cooling effect has been reported by several observational studies [11, 12, 13, 14, 15, 16].

(**SC3.10**) *P18 L1, Table 6 lists the mean CREs between... It should be "Table 6 lists the difference of the mean CREs between...". The same corrections are necessary for the following sentences.*

We thank the referee for highlighting the mistake. The text has been revised.

(**SC3.11**) *P18 L8. For a given liquid water path, the smaller... Check if it is grammatically correct.*

We double-checked the sentence and think it is correct.

(**SC3.12**) *P18 L16, a Pearson correlation of 0.950 (0.928) is yielded. The values are 0.952 (0.930) in Table 6.*

We thank the referee for highlighting the mismatches between the table and the text. Nevertheless, we now extended the analysis over all days and, thus, tables and related text have been revised.

(**SC3.13**) *P18 L19, a Pearson correlation of 0.995. The value is 0.996 in Table 6.*

The same as in (SC3.12).

(**SC3.14**) *P18 L19, and P20 L13, no suprise considering surprise*

The text has been revised as follows:

The latter can be explained by the way the droplet number concentration is derived (see Eq. 4)...

(**SC3.15**) *P18 L25, of about -6.52 $Wm^{-2}$ with a RMSE of 10.4 $Wm^{-2}$ for b and -9.31 $Wm^{-2}$ with a RMSE of 19.4 $Wm^{-2}$ for d The sign of -6.52 and -9.31 is different from that in Table 6. The mismatch is also in the number of -0.11 and -3.64 at Page 19 Line 1.*

We thank the referee for highlighting the mismatches between the table and the text. We have now extended the analysis over all days and, thus, tables and related text have been revised.

(**SC3.16**) *Table 6. The names of Scen. are wrong.*

The referee is correct. However, we now have decided to replace the sub-scenario (d, clusters) with a new scenario 4, whereby we employ the mean droplet number concentration profile over all days. Accordingly, Table 6 has been revised. The referee is referred to the section **General changes**.

(**SC3.17**) *P19 L3, For instance, in case of the adiabatic scenarios... the sub-adiabatic.*

The text is revised accordingly.

(**SC3.18**) *P19 L7, slightly larger scatter is found for S4 as compared to S3. Why is the result of S4 worth than that of S3?*

The difference between the sub-adiabatic model (S3) and the modified one (S4) is that the latter accounts for the depletion of the liquid water content due to entrainment, precipitation, and freezing drops. Consequently, we wanted to check whether it captures better the vertical stratification of the modeled low-level clouds and, accordingly, if it approximates the CREs of the reference simulation with better accuracy. Since S4 does not provide any further insight, we now have decided to drop this scenario. However, we do confirm that, by considering all the case days into the analysis, we came to the same conclusions as for 3 June. As a confirmation, we updated the Tables and attached them at the end of this document. The referee is referred to Tables R1–R3.

(**SC3.19**) *P20 L7: -0.76. It has different significant figures from that in Table 7. Same for 0.21 at P20 L13.*

We thank the referee for highlighting the mismatches. We have now extended the analysis over all days and, thus, tables and related text have been revised.

(**SC3.20**) *Logarithmic axis is preferred. The saturation may not be found in the logarithmic plot.*

We acknowledge the referee's suggestion for the logarithmic axis. However, we respectfully disagree on this point. Firstly, we would like to highlight that, in the SW radiation, an excellent monotonic relation is found between the CREs and cloud optical thickness, liquid water path, and cloud geometrical extent for both BOA and TOA. Secondly, the SW CRE is negative and the logarithm of a negative number is undefined. Even if we take the absolute value of the CRE, we still see the monotonic relation, but, it is less pronounced.

(**SC3.21**) *P21 L3, e.g., Fig. 6 panels (a) or (b) with Fig. 8 panel (b) Fig. 8 panel (a).*

We thank the referee for pointing to the mistake. Nevertheless, we now have reduced the amount of plots. Following the general comment of the anonymous referee #2, we now illustrate only the results for TOA (see Figs. R3 and R4).

(**SC3.22**) *P21 L3, The resulting Spearman and Pearson correlations larger than 0.96 and 0.91, respectively. The values seem to be 0.816 and 0.914.*

The correct panels of Fig. 8 are (a) and (d). Accordingly, the Spearman correlation is 0.96 and 0.935. Nevertheless, we now extended the analysis over all case days and, thus, the correlations have slightly changed.

(**SC3.23**) *P21 L14, with Spearman and Pearson correlations above -0.796 and -0.82, respectively. The values should be -0.820 and -0.796. The author should mention that these values are only for high values of the droplet number concentration.*

The referee is correct. However, we now have decided to replace the sub-scenario (d, clusters) with a new scenario 4, whereby we employ the mean droplet number concentration profile over all case days. For details with respect to the relevant changes, the referee is referred to section **General changes**.

**List of Figures**

[revised manuscript text omitted]

Table R8: Mean and standard deviation of modeled CREs (W m$^{-2}$) for the SW, LW, and NET (SW + LW) radiation for the reference simulation over all case days. ATM stands for the atmospheric cloud radiative effect defined as the difference between the CREs at the TOA and BOA.

| Ref. | $CRE_{SW}$ | $CRE_{LW}$ | $CRE_{NET}$ |
|---|---|---|---|
| TOA | −348.7 ± 78.39 | 17.51 ± 10.04 | −331.2 ± 77.27 |
| ATM | 32.94 ± 12.11 | −39.16 ± 13.14 | −6.225 ± 12.98 |
| BOA | −381.6 ± 86.95 | 56.66 ± 9.746 | −324.9 ± 86.51 |

Table R9: Mean CRE $(\mathrm{W\,m^{-2}})$ for the SW radiation. Results are given as differences between the new scenario minus the reference simulation ($\Delta$). The root mean square error (RMSE) in $\mathrm{W\,m^{-2}}$ and the Pearson (Pears.) correlation between the new scenarios and the reference simulation are also given.

| Scen. | $\mathrm{CRE_{SW,B}}$ | | | $\mathrm{CRE_{SW,T}}$ | | |
|---|---|---|---|---|---|---|
| | $\Delta$ | RMSE | Pears. | $\Delta$ | RMSE | Pears. |
| S1a | $-39.2$ | 46.4 | 0.960 | $-40.1$ | 47.0 | 0.952 |
| S1b | $-7.04$ | 11.7 | 0.995 | $-6.53$ | 11.7 | 0.994 |
| S1c | $-2.59$ | 23.4 | 0.964 | $-1.86$ | 24.3 | 0.951 |
| S2a | $-26.1$ | 39.2 | 0.943 | $-27.1$ | 39.8 | 0.930 |
| S2b | 7.74 | 14.2 | 0.991 | 8.19 | 13.6 | 0.990 |
| S2c | 12.9 | 32.4 | 0.943 | 13.7 | 33.6 | 0.921 |
| S3a | $-31.1$ | 41.4 | 0.950 | $-32.9$ | 42.9 | 0.937 |
| S3b | 1.47 | 10.6 | 0.993 | 1.17 | 10.0 | 0.992 |
| S3c | 6.59 | 27.7 | 0.953 | 6.55 | 29.0 | 0.934 |
| S4 | $-3.13$ | 16.7 | 0.983 | $-3.16$ | 17.2 | 0.977 |

Table R10: Correlations between the cloud radiative effects for the reference simulation (Ref.) and the cloud properties. For the SW (LW) radiation, results are presented in case of the Spearman (Pearson) correlation.

| Properties | $\mathrm{CRE_{SW,B}}$ | $\mathrm{CRE_{SW,T}}$ | $\mathrm{CRE_{LW,B}}$ | $\mathrm{CRE_{LW,T}}$ |
|---|---|---|---|---|
| | Spearman | | Pearson | |
| $Q_\mathrm{L}$ | $-0.957$ | $-0.955$ | $-0.129$ | 0.181 |
| $\tau$ | $-0.994$ | $-0.987$ | 0.104 | 0.148 |
| $N_\mathrm{int}$ | $-0.471$ | $-0.431$ | 0.428 | $-0.290$ |
| $r_\mathrm{int}$ | $-0.446$ | $-0.460$ | $-0.395$ | 0.344 |
| CBH | 0.148 | 0.063 | $-0.389$ | 0.752 |
| CTH | 0.143 | $-0.220$ | $-0.428$ | 0.765 |
| $H$ | $-0.795$ | $-0.812$ | $-0.200$ | 0.226 |
| $f_\mathrm{ad}$ | $-0.284$ | $-0.273$ | 0.145 | 0.134 |

Table R11: Mean CRE ($\mathrm{W\,m^{-2}}$) for the LW radiation. Results are given as differences between the new scenario minus the reference simulation ($\Delta$). The root mean square error (RMSE) in $\mathrm{W\,m^{-2}}$ and the Pearson (Pears.) correlation between the new scenarios and the reference simulation are also given.

| Scen. | $\mathrm{CRE_{LW,B}}$ | | | $\mathrm{CRE_{LW,T}}$ | | |
|---|---|---|---|---|---|---|
| | $\Delta$ | RMSE | Pears. | $\Delta$ | RMSE | Pears. |
| S1a | $-0.11$ | 0.48 | 0.999 | $-0.04$ | 0.19 | 1.000 |
| S1b | $-0.05$ | 0.40 | 0.999 | $-0.03$ | 0.18 | 1.000 |
| S1c | $-0.01$ | 0.50 | 0.999 | $-0.01$ | 0.22 | 1.000 |
| S2a | 0.40 | 0.79 | 0.998 | 0.23 | 0.51 | 0.999 |
| S2b | 0.51 | 0.82 | 0.998 | 0.27 | 0.53 | 0.999 |
| S2c | 0.55 | 0.85 | 0.998 | 0.29 | 0.54 | 0.999 |
| S3a | $-0.05$ | 0.74 | 0.997 | 0.33 | 0.64 | 0.999 |
| S3b | $-0.01$ | 0.73 | 0.997 | 0.36 | 0.65 | 0.999 |
| S3c | 0.02 | 0.83 | 0.996 | 0.37 | 0.68 | 0.998 |
| S4 | $-0.02$ | 0.49 | 0.999 | $-0.02$ | 0.22 | 1.000 |

---

## Author Comment (AC2) · 8 Oct 2019

**Answers to Anonymous Referee #1**

We thank the anonymous referee #1 for his/her constructive comments and suggestions that certainly have improved the manuscript significantly. We revised the manuscript according to his/her comments and the comments of anonymous referees #2 and #3. In the following,

- *referee's comments are given in italic,*
- our answers are outlined in normal format, and
- **textual changes in the manuscript are given in bold format**.

We would like inform the anonymous referee #1 about the following changes:

1. Driven by the specific comment (SC) #18 of anonymous referee #3 (SC3.18), we decided to drop scenario S4 from the analysis. The difference between the sub-adiabatic model (S3) and the modified one (S4) is that the latter accounts for the depletion of the liquid water content due to entrainment, precipitation, and freezing drops. Consequently, we wanted to check whether S4 captures better the vertical stratification of the modeled low-level clouds and, accordingly, if it approximates the CREs of the reference simulation with better accuracy. Since S4 does not provide any further insight, we now have decided to drop this scenario. However, we do confirm that, by considering all the case days in the analysis, we came to the same conclusions as for 3 June. As a confirmation, we updated the Tables and attached them at the end of this document. The referee is referred to Tables R1–R3.

2. In all scenarios, we decided to drop sub-case d, which employs two fixed values for the droplet number concentration representing the two modes in the corresponding histogram for 3 June 2016. This scenario separates clouds into a cluster with low/high clouds. Considering the vertical variability of the droplet number concentration, the latter clustering will link low clouds (within the boundary layer) with high $N_\mathrm{d}$ and, accordingly, high clouds with lower $N_\mathrm{d}$ values. Thus, for all scenarios, employing such values for $N_\mathrm{d}$ are able to approximate the reference radiative transfer simulation very well. Only the radiative transfer simulation that is supplied by the droplet number concentration weighted over the cloud geometrical extent, i.e., $N_\mathrm{int}$ (sub-case b) leads to smaller differences when compared to the reference simulation. However, we do confirm that, by considering all the case days into the analysis, we came to the same conclusions as for 3 June. Note that, for the latter case, the clustering was conducted on the mean $N_\mathrm{int}$ over all case days. As a confirmation, we updated the Tables and attached them at the end of this document. The referee is referred to Tables R1–R3.

3. We decided to add a new scenario as a replacement of sub-case d, whereby radiative transfer simulations are conducted for a mean vertical profile of the droplet number concentration over all case days. Tables R9–R11 summarize the new results. In brief, this scenario is considered as an improvement compared to the clustering case. The following parts were included within the text:

   Section 5.1.2: **Last but not least, by replacing the vertical profile of $N_\mathrm{d}$ by the**

mean profile of $N_{\rm d}$ over all case days (see Fig. 2), emulates the cloud radiative effects of the reference simulation quite well. Accordinly, scenario S4 slightly undersimates the mean SW CREs, with an mean error up to $-3.16\,\rm W\,m^{-2}$ and a RMSE up to $17.2\,\rm W\,m^{-2}$ for both BOA and TOA. In fact, this scenario outperforms the rest scenarios (S1–S3), except from the sub-case b ($N_{\rm int}$) in all scenarios. For an illustration of the excellent linear correlation between the reference simulation and S4 by means of a bivariate kernel density (BKD) plot, the reader is referred to Fig. B1 in Appendix B. One can see that the CREs computed by these scenarios are in a very good agreement almost everywhere except towards larger values of the CREs in case of the SW radiation, with Pearson correlations larger than 0.977 for both BOA and TOA.

Section 6: **By employing a more representative profile for the $N_{\rm d}$, i.e., a mean vertical profile of $N_{\rm d}$ over all case days leads to a rather good approximation; the RMSE is below $17.2\,\rm W\,m^{-2}$. This points to the need to better account for prognostic $N_{\rm d}$ calculations.**

Appendix B: **In sect. 5.1.2, by conducting idealized radiative transfer simulations, we estimated the impact of the representation of cloud properties in ICON-LEM on the cloud radiative effects (CREs). Special emphasis was given on identifying the droplet number concentration ($N_{\rm d}$), which approximates the microphysical and radiative properties of low-level clouds as simulated by ICON-LEM (reference scenario). A radiative transfer simulation, which employs a mean vertical profile of $N_{\rm d}$ over all the case days (scenario S4), approximates the CREs of the reference scenario quite well. Figure B1 depicts the excellent linear correlation between the reference simulation and S4 by means of a bivariate kernel density (BKD).**

4. Following the general comment of anonymous referee #2 for shortening the manuscript given the redundancy of many of the results shown in this study and his/her relevant specific comments (SC), i.e., (SC2.12) and (SC2.25):

   - We decided to drop Fig. B1. Figure B1 illustrates the bivariate kernel density (BKD) between the cloud optical thickness and the liquid water path on a logarithmic scale. Considering the comprehensive explanation given in Sect. 3.3.1, we decided that this illustration did not provide any additional information.

   - Figures 6 and 7 have been revised. Now, they illustrate results only for TOA (see Figs R2 and R3).

   - We now focus only on the rotational component analysis. The mention of the principal component analysis have been significantly reduced. In addition, we removed the relevant information from Table 3. For the updated version of the Table, the referees are referred to Table R5. Additionally, we replaced Figure 5 by Table R4. This table lists the contribution of each rotational component to the total variance.

**Answers to general comments (GC) from referee #1 (GC1)**

**(GC1.1**) *My major concern is that the simulations examined in this study are very limited (Page 8 Line 15-18). The authors conducted simulations using six case days, but actually looked at in details only the case of 3 June 2016. How general are they? Doesn't the vertical structure of adiabaticity depend strongly on the cloud regimes and types or their life-stage? In the present form of this paper, objectives are too narrow. The described relationship among cloud micro- and macrophysical properties and radiative effect using high resolution simulation may provide key suggestions on aerosol-cloud interactions, but the findings as they are, are by no means general. With some more simulation cases or a bit more analysis for all the case days in detail, I think this will make a publishable work.*

These days have been selected from the total set of available case days by the presence of suitable liquid water cloud fields and no known bugs in the used model version, which affect the representation of low-level clouds. We do agree that the vertical structure of adiabaticity depends on cloud regimes, types, and life-stage and, thus, it could be an interesting extension. However, due to the high horizontal resolution of ICON-LEM, for a single day, the number of "independent" cloudy columns are very large and complicates the investigation of such dependencies. Note here that the model output employed in this study, 3D HOPE data, has an output frequency of 15 min, while the domain size is limited to $45\,km^2$. For such studies, especially when it comes to life-stage, it would be better to use model data with higher output frequency, e.g., 1D profiles that are available every 10 sec. But, this is beyond the purpose of this study. However, we revised our manuscript according to the comments of anonymous referee #1 and the comments of anonymous referees #3 further extended our analysis over all days to improve the robustness of our results. Now, sections 3.2 and 5 outline our findings for all case days.

As we aforementioned, throughout this study, a special emphasis was given to 3 June 2016, because, regardless of the large variability in cloud properties for each day, it approximates best the mean properties over all the case days considered. Thus, the revision of these two plots did not require any significant textual alteration (see Fig. R1 and Fig. R2); only minor textual changes were made.

**Answers to specific comments (SC) from referee #1 (SC1)**

**(SC1.1**) *Section 2.3: Please describe the model resolution, domain size, as well as timestep used in the simulations. The general description of ICON-LEM on page 3 (L16-17 and L28-30) is confusing with regards to this.*

We revised the description of ICON-LEM (Section 2.1) according to this comment and the specific comment (SC) # 3 of anonymous referee #3 (SC3.3):

The ICON unified modeling framework was co-developed by the German meteorological service (DWD) and the Max Planck institute for meteorology (MPI-M) in order to support climate research and weather forecasting. Within the HD(CP)2 project, ICON was further extended towards large eddy simulations with realistic topography **and** open boundary conditions. This resulted in

ICON-LEM deployed in restricted areas that are centered on Germany and the Tropical Atlantic [1]. The equations utilized by the model are based on the prognostic variables given by Gassmann and Herzog [2]. These variables comprise the horizontal and vertical velocity components, the density of moist air, the virtual potential temperature, and the mass and number densities of traces, e.g., specific humidity, liquid water, and different ice hydrometeors. A comprehensive description of the model and its governing equations is found in Dipankar et al. [3] and Wan et al., [4]. Concerning turbulence parameterization, the three-dimensional Smagorinsky scheme is employed [3]. The activation of cloud condensation nuclei (CCN) is based on the parameterization of Seifert and Beheng [5] and modified in order to account for the consumption of CCNs due to their activation into cloud droplets. The CCN concentration is then parameterized following the pressure profile and the vertical velocity [6].

Simulations are carried out for three different domains with 624 m, 312 m, and 156 m horizontal resolution. The model domains consist of 150 vertical levels, with resolutions ranging from ~25 m to 70 m within the boundary layer, and from 70 m to 355 m further up until the top of the domain at 21 km. For each of the aforementioned grids, **data** is stored as one-dimensional (1D) profiles every 10 sec, two-(2D), and 3D snapshots [1]. The model yields output on each of the aforementioned grids with the data stored as one-dimensional (1D) profiles, two-(2D), and 3D snapshots [1]. In case of the 3D output, the simulation data is interpolated from the original grids (e.g., 156 m) to a 1 km grid, the 3D coarse data, and 300 m grid, the so-called HOPE data. The latter output has been created for the purpose of model evaluation with ground-based observations from the $HD(CP)^2$ Observational Prototype Experiment (HOPE) that took place near Jülich [7]**and is limited to a domain size of about ~45 km². Note here that for the 2D and 3D output, data is stored at day- and night-time frequency. Day-time frequency begins at 06:00 UTC and lasts until 00:00 UTC, while night-time starts at midnight and lasts until 06:00 UTC. The 2D data is stored with a day-time and night-time frequency of 10 sec and 5 min, respectively. The 3D coarse data have day-time frequency of 10 min (1 hour at night-time). In this study, the 3D HOPE data has been used that is stored only at a day-time frequency of 15 min.**

(**SC1.2**) *Equation (7): It is better to add a sentence about the factor* 2/3, *rather than* 5/9, *citing relevant papers (e.g., Szczodrak et al., 2001; Wood and Hartmann, 2006; Lebsock and Su, 2014). Equations (14) and (15) as well.*

We revised section 2.6 according to this comment and the specific comment (SC) #4 of anonymous referee #3 (SC3.4). The following parts have been added:

P6 L24: **while the factor** 2/3 **is a scale factor resulting from the constant liquid water content and effective radius with height [8].**

P7 L21: **Compared to Eq. (7), Eq. (10) leads to a factor of** 5/9**, meaning that the sub-adiabatic liquid water path is** 5/6 **times the one of the vertically homogeneous model [9].**

P8 L11: **For vertically constant** $q_L$ **and** $r_{eff}$**, this can be interpreted as the cloud optical thickness coming from the vertical homogeneous model (see Eq. 7). According**

to the sub-adiabatic cloud model, the cloud optical thickness is linked to the liquid water path and the effective radius [10],

$$\tau = \frac{9}{5} \frac{Q_{\mathrm{L}}}{\rho_{\mathrm{w}} \cdot r_{\mathrm{eff}}}$$

Alternatively, substituting $r_{\mathrm{eff}}$ from Eq. (13) in Eq. (15), the cloud optical thickness is given by,...

(**SC1.3**) *Figure 3 and caption: $q_{\mathrm{L}}$ -¿ $Q_{\mathrm{L}}$ or CLWP.*

Actually, Fig. 3 illustrates the mean liquid water content profile normalized over the cloud geometrical extent. Throughout the paper, liquid water content is denoted as $q_{\mathrm{L}}$.

(**SC1.4**) *Figures 6, 7 and 8: The order of sub-figures is not consistent with the caption.*

The order of sub-figures has been revised for consistency.

(**SC1.5**) *Table 6: I found several mismatches between Table 6 and citing main text (e.g., P18 L16), which made reviewers very difficult to track...*

We apologize for the mismatches. We have now extended the analysis to all case days and, thus, tables and related text have been revised.

(**SC1.6**) *P23 L11-13: This sentence is too vague. Please raise more specific source of uncertainty, and describe how the scrutinization is required.*

After the additional insight given by anonymous referee #2 (see general comment 3, i.e., GC2.3) we have revised this part of the text as follows:

The vertical variability of the droplet number concentration was examined. **For all the case days**, above an altitude of **about** 2 km, values of $N_{\mathrm{d}}$ are about $200\,\mathrm{cm}^{-3}$ and are, thus, close to climatological values, while in the boundary layer, the double moment scheme predicts $N_{\mathrm{d}}$ values above $600\,\mathrm{cm}^{-3}$. Such values are **regarded as** rather high compared to satellite remote sensing estimates [11, 12], **but such comparison is rather vague considering, firstly, the large uncertainties of the satellite-derived estimates of cloud droplet number concentration [12] and, secondly, they are not available in high resolution. However,** in situ observations, **which are considered to be the most accurate approach to determine $N_{\mathrm{d}}$,** suggest higher values and, hence, lie closer to those simulated by ICON-LEM. **Thus, by means of in situ observations, evaluation activities should be conducted for a better characterization of the droplet number concentration from remote sensing techniques. The latter will scrutinize the double-moment scheme implemented in ICON-LEM and could potentially lead to better simulations of cloud processes and radiation.**

We additionally revised the corresponding text in Section 3.2 as follows,

**On the contrary**, in situ observations suggest higher values of $N_{\mathrm{d}}$ and, accordingly, closer to those simulated by ICON-LEM. **Hence, efforts should be undertaken to further validate**

**the cloud droplet number concentrations predicted by the double-moment scheme.**

**(SC1.7**) *P25 L12-L13: This sentence recommends double-moment cloud microphysics, but P23 L12 points weakness of the double-moment.*

After the insight given by the anonymous referee #2, the sentence at page 23, line 12, has been revised (see SC.6).

**(SC1.8**) *Appendix B: Please change the appendix title. Appendix section is not just a list of supporting materials. The current version does not have any explanation about the figures in the appendix (Appendix C as well).*

The Appendix B and C have been revised.

**List of Figures**

[revised manuscript text omitted]

---

## Author Comment (AC3) · 8 Oct 2019

**Answers to Anonymous Referee #2**

We thank the anonymous referee #2 for his/her constructive comments and suggestions that certainly have improved the manuscript significantly. We revised the manuscript according to his/her comments and the comments of anonymous referees #1 and #3. In the following,

- *referee's comments are given in italic,*

- our answers are outlined in normal format, and

- **textual changes in the manuscript are given in bold format**.

**General changes**

We would like inform the anonymous referee #2 about the following changes:

1. Driven by the specific comment (SC) #18 of anonymous referee #3 (SC3.18), we decided to drop scenario S4 from the analysis. The difference between the sub-adiabatic model (S3) and the modified one (S4) is that the latter accounts for the depletion of the liquid water content due to entrainment, precipitation, and freezing drops. Consequently, we wanted to check whether S4 captures better the vertical stratification of the modeled low-level clouds and, accordingly, if it approximates the CREs of the reference simulation with better accuracy. Since S4 does not provide any further insight, we now have decided to drop this scenario. However, we do confirm that, by considering all the case days in the analysis, we came to the same conclusions as for 3 June. As a confirmation, we updated the Tables and attached them at the end of this document. The referee is referred to Tables R1–R3.

2. In all scenarios, we decided to drop sub-case d, which employs two fixed values for the droplet number concentration representing the two modes in the corresponding histogram for 3 June 2016. This scenario separates clouds into a cluster with low/high clouds. Considering the vertical variability of the droplet number concentration, the latter clustering will link low clouds (within the boundary layer) with high $N_\mathrm{d}$ and, accordingly, high clouds with lower $N_\mathrm{d}$ values. Thus, for all scenarios, employing such values for $N_\mathrm{d}$ are able to approximate the reference radiative transfer simulation very well. Only the radiative transfer simulation that is supplied by the droplet number concentration weighted over the cloud geometrical extent, i.e., $N_\mathrm{int}$ (sub-case b) leads to smaller differences when compared to the reference simulation. However, we do confirm that, by considering all the case days into the analysis, we came to the same conclusions as for 3 June. Note that, for the latter case, the clustering was conducted on the mean $N_\mathrm{int}$ over all case days. As a confirmation, we updated the Tables and attached them at the end of this document. The referee is referred to Tables R1–R3.

3. We decided to add a new scenario as a replacement of sub-case d, whereby radiative transfer simulations are conducted for a mean vertical profile of the droplet number concentration over all case days. Tables R9–R11 summarize the new results. In brief, this scenario is considered as an improvement compared to the clustering case. The following parts were included within the text:

Section 5.1.2: **Last but not least, by replacing the vertical profile of $N_{\mathrm{d}}$ by the mean profile of $N_{\mathrm{d}}$ over all case days (see Fig. 2), emulates the cloud radiative effects of the reference simulation quite well. Accordinly, scenario S4 slightly undersimates the mean SW CREs, with an mean error up to $-3.16\,\mathrm{W\,m^{-2}}$ and a RMSE up to $17.2\,\mathrm{W\,m^{-2}}$ for both BOA and TOA. In fact, this scenario outperforms the rest scenarios (S1–S3), except from the sub-case b ($N_{\mathrm{int}}$) in all scenarios. For an illustration of the excellent linear correlation between the reference simulation and S4 by means of a bivariate kernel density (BKD) plot, the reader is referred to Fig. B1 in Appendix B. One can see that the CREs computed by these scenarios are in a very good agreement almost everywhere except towards larger values of the CREs in case of the SW radiation, with Pearson correlations larger than 0.977 for both BOA and TOA.**

Section 6: **By employing a more representative profile for the $N_{\mathrm{d}}$, i.e., a mean vertical profile of $N_{\mathrm{d}}$ over all case days leads to a rather good approximation; the RMSE is below $17.2\,\mathrm{W\,m^{-2}}$. This points to the need to better account for prognostic $N_{\mathrm{d}}$ calculations.**

Appendix B: **In sect. 5.1.2, by conducting idealized radiative transfer simulations, we estimated the impact of the representation of cloud properties in ICON-LEM on the cloud radiative effects (CREs). Special emphasis was given on identifying the droplet number concentration ($N_{\mathrm{d}}$), which approximates the microphysical and radiative properties of low-level clouds as simulated by ICON-LEM (reference scenario). A radiative transfer simulation, which employs a mean vertical profile of $N_{\mathrm{d}}$ over all the case days (scenario S4), approximates the CREs of the reference scenario quite well. Figure B1 depicts the excellent linear correlation between the reference simulation and S4 by means of a bivariate kernel density (BKD).**

4. Following the general comment of anonymous referee #2 for shortening the manuscript given the redundancy of many of the results shown in this study and his/her relevant specific comments (SC), i.e., (SC2.12) and (SC2.25):

   - We decided to drop Fig. B1. Figure B1 illustrates the bivariate kernel density (BKD) between the cloud optical thickness and the liquid water path on a logarithmic scale. Considering the comprehensive explanation given in Sect. 3.3.1, we decided that this illustration did not provide any additional information.

   - Figures 6 and 7 have been revised. Now, they illustrate results only for TOA (see Figs R2 and R3).

   - We now focus only on the rotational component analysis. The mention of the principal component analysis have been significantly reduced. In addition, we removed the relevant information from Table 3. For the updated version of the Table, the referees are referred to Table R5. Additionally, we replaced Figure 5 by Table R4. This table lists the contribution of each rotational component to the total variance.

**Answers to general comments (GC) from referee #2 (GC2)**

**(GC2.1)** *There is already a significant body of work on the topic of sub-adiabaticity (much of which the authors cite) and the results of this study seem to confirm past findings (in particular, those of Merk et al., 2016) with little added insight into process (radiative, microphysical, etc.) besides pointing out that single-moment microphysics schemes leave much to be desired (which has been explored by e.g., Igel et al., 2015, JAS). Much work has also been done with respect to statistical emulators for understanding cloud radiative effects (e.g. Feingold et al., 2016; Glassmeier et al., 2019) and the aggregation of model data over all shallow cloudy columns severely limited the authors' ability to examine details regarding differences between cumulus and stratus (which likely exhibit very different fad), the diurnal cycle, or radiative effects across spatial scales – an exploration of the latter would be especially useful since the 300 m HOPE dataset is finer in horizontal resolution than the existing remote sensing products this study is designed to improve (typically 1 km pixel size).*

We acknowledge Referee's #2 concerns with respect to the novelty of this study. However, we respectfully disagree on this point. ICON-LEM domain consists of 150 vertical levels, with resolutions ranging from 25 m to 70 m within the boundary layer, from 70 m to 100 m further up to the altitude limit for the occurrence of low-level clouds selected for this study (4000 m), and from 70 m to 355 m further up until the top of the model domain at 21 km. This unprecedented high vertical resolution enables a significantly improved investigation of the vertical distribution of microphysical properties of low-level clouds as simulated by a double-moment scheme. We do agree that the vertical structure of adiabaticity depends on cloud regimes, types, and life-stage and, thus, it could be an interesting extension. However, due to the high horizontal resolution of ICON-LEM, for a single day, the number of "independent" cloudy columns are very large and complicates the investigation of such dependencies. Note here that the model output employed in this study, 3D HOPE data, has an output frequency of 15 min, while the domain size is limited to 45 km$^2$. For such studies, especially when it comes to life-stage, it would be better to use model data with higher output frequency, e.g., 1D profiles that are available every 10 sec. But, this is beyond the purpose of this study.

However, we revised our manuscript according to the comments of anonymous referees #1 and #3 and further extended our analysis to consider all case days to improve the robustness of our results. Now, sections 3.2 and 5 outline our findings for all case days.

**(GC2.2)** *Finally, I get the sense that this paper only deals with sub-adiabaticity in passing – the latter half of the paper is primarily concerned with describing CREs with a minimal set of variables and is almost completely disconnected from the title of the paper. Sub-adiabaticity seems to have only a weak influence on CREs.*

A high-resolution model as ICON-LEM is an ideal tool to investigate the suitability of the sub-adiabatic cloud model, firstly, for the evaluation of the representation of low-level clouds and, secondly, to capture the relevant properties which determine the cloud radiative effect. This outlines our main objectives and we think that it is reflected by the title of the paper.

We do not completely agree that the sub-adiabatic fraction has only a weak influence on the cloud radiative effect (CRE). In the first place, the sub-adiabatic fraction is the key component for

deriving the cloud optical thickness that is one of the fundamental cloud properties for describing the shortwave (SW) cloud radiative effects (CREs). Based on six case days, we found that the behavior of modeled liquid water clouds over Germany more closely resembles the sub-adiabatic model than the vertically homogeneous one, with a mean sub-adiabatic fraction ($f_{ad}$) of about 0.45. This model suggests, e.g., scaling of $\log(\tau)/\log(Q_L)$ with 5/6 and $f_{ad} < 1$. This scaling behavior has implications to, at least, the shortwave (SW) CRE. In addition, Eq. (15) contains the factor $\tau \propto f_{ad}^{-1/6}$. The latter factor, in combination with the mean sub-adiabatic fraction found in this study has a significant impact in $\tau$ compared to the pure adiabatic assumption that is usually employed.

Last but not least, the rotational component analysis (principal component and varimax rotation), clearly identifies the sub-adiabatic fraction as one of the minimal set of parameters to explain the CREs. In fact, it shows up as the 3$^{rd}$ rotational component (RC-3) that explains 14.8 % of the total variance.

(GC2.3) *With respect to motivation, the authors rely heavily and repeatedly on the idea that there are large uncertainties in aircraft measurements of cloud drop number concentration ($N_d$ in the authors' notation), which they justify by citing the $N_d$ retrieval review paper of Grosvenor et al. (2018) – specifically, I believe they refer to Grosvenor et al.'s Figure 5 (which is in turn based on data used in Siebert et al., 2013) and accompanying discussion. This is an unfortunate figure. The disagreement of two probes (Phase Doppler Interferometer and Particulate Volume Monitor; PDI and PVM, respectively) at concentrations of $N_d > 350\,cm^{-3}$ is used as evidence that in situ probes have a general, systematic problem measuring $N_d$.*

*The issue with this illustration is that one of the two probes used (PVM) is not designed to measure $N_d$ and I am aware of no other publication in which this is even attempted. The PVM measures extinction from a population of cloud drops and makes no explicit count of particle density. In fact, I'm not even sure how this quantity was generated since the PVM returns only two data streams: total particle volume and surface area. The PDI, on the other hand, is frequently used by both the airborne cloud physics and industrial spray characterization communities and has been demonstrated to accurately count (and size) particles up to a concentration of O(105) cm-3. An intercomparison of PDI with other probes that explicitly count particles (CAS, FSSP, CDP, Holodec. . . etc. – there are a great number and I don't understand why Grosvenor et al. chose such an ill-suited probe for their figure) would likely show a much better overlap in the PDFs of $N_d$ from different probes; such an intercomparison of the latest generation of cloud probes is currently underway for the recent NASA ORACLES campaign, which sampled a wide variety of concentration conditions due to the campaign's focus on interaction of clouds with overlying smoke layers during the stratocumulus to cumulus transition.*

*I am strongly opposed to the use of phrasing such as "large instrumental uncertainties" (e.g. page 23, lines 10-11) as I think this point is vastly overstated by Grosvenor et al. (2018), an assertion backed by their discussion of myriad other issues with retrieval assumptions ahead of any problems with in situ measurements.*

We thank the anonymous referee #2 for the insight given. We revised this part of the text as follows:

The vertical variability of the droplet number concentration was examined. **For all the case days**, above an altitude of **about** $2 \, \mathrm{km}$, values of $N_\mathrm{d}$ are about $200 \, \mathrm{cm}^{-3}$ and are, thus, close to climatological values, while in the boundary layer, the double moment scheme predicts $N_\mathrm{d}$ values above $600 \, \mathrm{cm}^{-3}$. Such values are **regarded as** rather high compared to satellite remote sensing estimates [1, 2], **but such comparison is rather vague considering, firstly, the large uncertainties of the satellite-derived estimates of cloud droplet number concentration [2] and, secondly, they are not available in high resolution. However,** in situ observations, **which are considered to be the most accurate approach to determine** $N_\mathrm{d}$**,** suggest higher values and, hence, lie closer to those simulated by ICON-LEM. **Thus, by means of in situ observations, evaluation activities should be conducted for a better characterization of the droplet number concentration from remote sensing techniques. The latter will scrutinize the double-moment scheme implemented in ICON-LEM and could potentially lead to better simulations of cloud processes and radiation.**

We additionally revised the corresponding text in Section 3.2 as follows,

**On the contrary**, in situ observations suggest higher values of $N_\mathrm{d}$ and, accordingly, closer to those simulated by ICON-LEM. **Hence, efforts should be undertaken to further validate the cloud droplet number concentrations predicted by the double-moment scheme.**

**Answers to specific comments (SC) from referee #2 (SC2)**

(**SC2.1**) *P2, L7: "taking placed" should be "taking place"*

The text is corrected.

(**SC2.2**) *P2, L21-22: "fixed droplet number distribution" – ambiguous terminology; "fixed droplet size distribution" would be clearer.*

The text is revised.

(**SC2.3**) *P2, L23: "Double-moment microphysical schemes. . . are only recently becoming more widespread": Perhaps in the operational forecasting community this is true, but in research modeling (especially of warm clouds), double-moment schemes have been common for at least a decade.*

The referee is correct. We revised the text by adding at the end of the sentence: **in operational forecasting**.

(**SC2.4**) *P5, L16: Why do you use an indirect measure for rain/drizzle instead of directly examining rain water mixing ratio? I understand that it makes for a more straightforward comparison with observations, but it seems like an unnecessary step.*

The referee is correct. The reasoning was to perform a straightforward link to observations. However, we do consider the rain water content as an additional threshold. Relevant information

has been included.

(**SC2.5**) *P6, L2: "The model outputs the..."*

The text is revised.

(**SC2.6**) *Is the assumption of vertical homogeneity a "scheme?" Seems like an odd word choice.*

The text is revised accordingly and the word "scheme" is replaced by "model".

(**SC2.7**) *P7, L15: "Clapeyron relationship"*

The word "relationship" has been included.

(**SC2.8**) *P7, L16-17: This sentence is difficult to follow. Rephrase and simplify the structure for clarity.*

The text is revised as follows: **For low level clouds, $\Gamma_{\mathrm{ad}}$ varies slightly ($\sim$20%). Consequently, in most studies, $\Gamma_{\mathrm{ad}}$ is assumed constant (e.g., Albrecht et al., 1990; Boers et al., 2006) or it is calculated from cloud bottom temperature and pressure (e.g., Merk et al., 2016) or cloud top information (e.g., Zeng et al., 2014).**

(**SC2.9**) *P11, L8: Remove "the" from "the 5 May..."*

The text is corrected.

(**SC2.10**) *P13, L12-13: "with a 5/6 slope" – possibly remove the word "fit," doesn't make sense in context.*

The text is revised accordingly.

(**SC2.11**) *P13, L18: If $f_{\mathrm{ad}}$ only accounts for 0.14% of the variance in $\tau$, what's the point of all this?*

Actually, in Sect. 3.3.1, we try to predict the cloud optical thickness derived from the output of ICON-LEM (by using Eq. 14), via employing the relevant equation suggested by the sub-adiabatic model, i.e., Eq. (15). Note here that, based on 6 case days, $f_{\mathrm{ad}}$ is 0.45 on average and not 1. For further information with respect to the relative importance of the sub-adiabatic fraction, the referee is referred to our answer at (GC2.2). This section has been revised:

Correction: $f_{\mathrm{ad}}$ accounts for 0.2 % of the variance in $\tau$.

**With this intention, an effort has been conducted to predict the cloud optical thickness derived from Eq. (14) by employing the sub-adiabatic model and Eq. (15).**

**In fact, model $Y_4(Q_{\mathrm{L}}, f_{\mathrm{ad}}, N_{\mathrm{int}})$ supports the applicability of the sub-adiabatic model since it is able to approximate the cloud optical thickness with high accuracy (RMSE =**

0.027)

(**SC2.12**) *P13, Section 4: The step by step narrative of the PC analysis is overwrought. If you primarily intend to use the results of the RC analysis to justify the minimal set of variables needed to represent CREs, skip the PC discussion; the PC and RC results are sufficiently similar that it is redundant.*

We understand the referee's concerns, but we do not entirely agree that PC and RC results are sufficiently similar. Although each PC is clearly dominated by some properties, they are found moderately or strongly correlated with the remaining properties. On the contrary, the rotational component analysis points to exactly which properties dominate at each RC. However, we do agree that we provided a comprehensive analysis and, hence, we decided to revise and shorten the text. We now focus only on the rotational component analysis. The mention of the principal component analysis have been significantly reduced. In addition, we removed the relevant information from Table 3. For the updated version of the Table, the referee is referred to Table R5. Additionally, we replaced Figure 5 by Table R4. This table lists the contribution of each rotational component to the total variance.

(**SC2.13**) *P13, L34: "optimal" instead of "optimized"*

The text is corrected.

(**SC2.14**) *P14, Table 3 caption: Remove trailing zero from "moderate [0.40, 0.6)" for consistency*

The text is revised accordingly.

(**SC2.15**) *P14, L7-9: Rearrange sentence beginning "However, the PCs..." to simplify structure for clarity.*

The text is revised: However, the PCs are hard to interpret. **Although each new dimension is clearly dominated by some of the cloud properties, the PCs are found moderately or strongly correlated with other properties.**

(**SC2.16**) *P14, L11: remove "so-called" – this makes it look like other people have a different name for it.*

The word "so-called" has been removed.

(**SC2.17**) *P15, L9-10: I am confused by what you're doing here – are you always running multiple simulations, or for scenarios S2-S4 are you imposing LWC/ND profiles that are not actually from the simulations?*

We only conduct radiative transfer simulations. For the reference scenario, the input for the RRTMG was constructed on the basis of ICON-LEM. In other words, temperature, pressure, and water vapour profiles, surface temperature and pressure, and cloud liquid water content and droplet number concentration. In the rest scenarios (S1-S4), we preserve the liquid water content and the $k_2$ parameter (taken from ICON-LEM) and we vary only the droplet number concentration. In

addition, for scenarios S2-S4, the liquid water path for each profile is re-distributed over the vertical. In this way, we can estimate the effects of the bulk microphysical parameterizations and the vertical stratification of the cloud properties on the CREs. (The relevant information is found in sections 5.1 and 5.1.1).

(**SC2.18**) *P15-16, L33-3: the assumptions would be more clearly expressed in a table.*

A table listing the details of all the assumptions has been added according the referee's suggestion (see Table R6).

(**SC2.19**) *P16, L1-2: Why are there drops in the free troposphere?.*

It is true that aerosols and their precursor gases are mostly produced in the boundary layer. However, they can be transported into the free troposphere via different mechanisms, such as trough convection and frontal uplift. There, their lifetime is much longer due to less efficient dry deposition as compared to the boundary layer and, accordingly, they can facilitate long-distance transport [3]. For example, Kupiszewski et al., (2013), reported that plumes in air aloft, above the boundary layer, can be attributed to transport of polluted air, e.g., via biomass burning. Biomass burning produces heat and moisture and this further leads to buoyancy-forced vertical and horizontal circulations of air and advection of hot gases [4]. The latter process is the main reason for the rapid uplift of smoke particles that are known to be an efficient CCN. Over the last decades, several studies reported aerosols in the free troposphere [5, 6] and even investigated CCN production there [7, 8, 9].

(**SC2.20**) *P16, L3: "where the liquid water path is preserved"*

The sentence is rearranged to be more clear: **Two different scenarios are considered, where the liquid water path is preserved within the vertical column, but the water content profile is redistributed.**

(**SC2.21**) *P16, L8: "following the climatology of a coarse..." – you only use the ECHAM value. Is this representative of what all GCMs do? If not, the generalization doesn't work.*

The same droplet number concentration profile is adopted by the regional climate model REMO [10]. A similar climatology is employed by ICON-NWP, which is the global Numerical Weather Prediction (NWP) version of ICON model heinze2017. The only difference in ICON-NWP is that the droplet number concentration within the boundary layer is $200\,\mathrm{cm}^{-3}$ and not $220\,\mathrm{cm}^{-3}$ as in ECHAM and REMO. An example study, whereby the climatology of $N_\mathrm{d}$ implemented in ECHAM was compared to satellite retrieved $N_\mathrm{d}$, is the one by Quaas et al., (2006). They retrieved $N_\mathrm{d}$ from MODIS and showed slightly lower values as compared to ECHAM $N_\mathrm{d}$ values, but consistent land-sea contrast [1].

The following part has been included in Section 3.1:

Note here that this value is close to the fixed droplet number concentration profile suggested by single-moment microphysical schemes adopted by atmospheric **models, such as ECHAM [11] and ICON-NWP, which is the global Numerical Weather Prediction (NWP) ver-**

sion of the ICON model [12].

(**SC2.22**) *P17, Section 5.1.2: I found the latter half of this discussion to be very difficult to follow, especially the references to various scenarios by only a letter or number near the end of the section (i.e. last paragraph, P18).*

We feel sorry for any inconvenience caused. The text has been revised.

(**SC2.23**) *P18, Table 5 caption: Cosine SZA was just given in text (and will hopefully be put in a separate table of assumptions) – remove since redundant.*

A table listing the details of all the assumptions has been added according the referee's suggestion (see R6). Thus, we removed the aforementioned information from the caption of Table 5.

(**SC2.24**) *P18, L1: "and the rest of the simulated..."*

The text is revised accordingly.

(**SC2.25**) *P19, Table 6: Two things: 1) numbering of scenarios is off by one and 2) since BOA and TOA are almost always within 5% or 1 W/m2 of each other, can you just pick one and reduce the amount of information here? This table would be much more effective/digestible.*

We thank the referee for highlighting the mistake in the numbering. We decided to keep the results for both BOA and TOA. However, we now have reduced the amount of scenarios employed in this study, hopefully making the table and the analysis easier to follow. In brief, we dropped scenario S4 (the modified sub-adiabatic mode), the sub-scenario (d, clusters), and included a new scenario representing the mean droplet number concentration profile over all case days. For a comprehensive description of all the changes made, the referee is referred to section **General changes** of the current document.

(**SC2.26**) *P19, L11-14: You can test whether effective radius is outside the range. Is this an issue or isn't it?*

For all the scenarios, we inter-compared only columns with valid values for the effective radius. Thus, we revised the text as follows:

Note here that the RRTMG model is able to derive the radiative fluxes only for effective radius between $2.5\,\mu m$ and $60\,\mu m$. **For all scenarios, all columns with effective radius outside this range have been excluded.**

(**SC2.27**) *P20, Section 5.1.3: As with the PCA results, what is the point of showing both correlations? You almost exclusively discuss Spearman, so why not just show that?*

We understand the referee's concerns with respect to the use of both Spearman and Pearson correlations. The principal component analysis reveals systematic co-variations among the cloud properties. These components can be seen as a linear combination among the original properties and, hence, we employ the Pearson correlation to describe their relation. However, in Section

5.1.3, we describe the correlation between the cloud radiative effects and the cloud properties and the rotational components. In case of the SW radiation, Spearman correlation is the ideal metric to describe the monotonic relation between the CREs and the cloud optical thickness, liquid water path, and cloud geometrical extent (and, accordingly, RC-2). On the other hand, in the LW radiation, due to the linear relationship between the CREs and the cloud bottom and top heights (and, accordingly, RC-1), the right metric to describe their relation is the Pearson correlation. We decided to keep both correlations, but revised the text so that we highlight their importance.

(**SC2.28**) *P20, L7: Capitalize "Spearman"*

Corrected.

(**SC2.29**) *P24, L3: "uncover potential shortcomings in...models": you only compared the model to itself, so how did you uncover shortcomings? Do you mean LES vs. GCM? Beyond discussing single- vs. double-moment microphysics (an already well-known issue), what shortcomings did you uncover?*

That was a mistake. We revised the text as follows: **The goal was ultimately to uncover potential shortcoming in the representation of clouds towards the computation of the cloud radiative effects.**

(**SC2.30**) *P24, L11: quantify contributions of 3rd/4th components to total variance here.*

We thank the referee for highlighting that we omitted an explicit reference to the contribution of the 3rd and 4th components to the total variance. These two components are clearly a function of the sub-adiabatic factor and the droplet number concentration (P20, L19), respectively, pointing to two clear degrees of freedom. They account for 14.8 % and 13.6 %, respectively, outlining their importance in identifying the minimum set of parameters for the representation of low-level clouds towards the computation of the CREs. Accordingly, we included the missing information. In addition, we decided to replace Figure 5 by a Table, where we list the contribution of each rotational component to the total variance.

(**SC2.31**) *P24, L11: delete "so-called"*

The word "so-called" has been removed.

(**SC2.32**) *P25, L9: again, is the ECHAM climatological ND representative? Is it even backed by observations? You have not made a case for why this is a good number to use, besides the fact that a single GCM uses it.*

we addressed the latter issue in (SC.21).

(**SC2.33**) *P25, L10: How do two fixed values constitute a profile?*

The referee is correct. The use of the word "profile" for a constant droplet number concentration over the vertical can be misleading. We have removed the profile and replace it by the word "values" throughout the manuscript.

(**SC2.34**) *P27, Eq A13: is exponent in denominator a typo? $D^0=1$.*

Indeed there was a typo. The denominator is actually the zeroth moment of the droplet size distribution, which corresponds to the droplet number concentration. The text has been revised.

**List of Figures**

[revised manuscript text omitted]

Table R9: Mean CRE ($\text{W}\,\text{m}^{-2}$) for the SW radiation. Results are given as differences between the new scenario minus the reference simulation ($\Delta$). The root mean square error (RMSE) in $\text{W}\,\text{m}^{-2}$ and the Pearson (Pears.) correlation between the new scenarios and the reference simulation are also given.

| Scen. | $\text{CRE}_{\text{SW,B}}$ | | | $\text{CRE}_{\text{SW,T}}$ | | |
|---|---|---|---|---|---|---|
| | $\Delta$ | RMSE | Pears. | $\Delta$ | RMSE | Pears. |
| S1a | $-39.2$ | 46.4 | 0.960 | $-40.1$ | 47.0 | 0.952 |
| S1b | $-7.04$ | 11.7 | 0.995 | $-6.53$ | 11.7 | 0.994 |
| S1c | $-2.59$ | 23.4 | 0.964 | $-1.86$ | 24.3 | 0.951 |
| S2a | $-26.1$ | 39.2 | 0.943 | $-27.1$ | 39.8 | 0.930 |
| S2b | 7.74 | 14.2 | 0.991 | 8.19 | 13.6 | 0.990 |
| S2c | 12.9 | 32.4 | 0.943 | 13.7 | 33.6 | 0.921 |
| S3a | $-31.1$ | 41.4 | 0.950 | $-32.9$ | 42.9 | 0.937 |
| S3b | 1.47 | 10.6 | 0.993 | 1.17 | 10.0 | 0.992 |
| S3c | 6.59 | 27.7 | 0.953 | 6.55 | 29.0 | 0.934 |
| S4 | $-3.13$ | 16.7 | 0.983 | $-3.16$ | 17.2 | 0.977 |

Table R10: Correlations between the cloud radiative effects for the reference simulation (Ref.) and the cloud properties. For the SW (LW) radiation, results are presented in case of the Spearman (Pearson) correlation.

| Properties | $\text{CRE}_{\text{SW,B}}$ | $\text{CRE}_{\text{SW,T}}$ | $\text{CRE}_{\text{LW,B}}$ | $\text{CRE}_{\text{LW,T}}$ |
|---|---|---|---|---|
| | Spearman | | Pearson | |
| $Q_{\text{L}}$ | $-0.957$ | $-0.955$ | $-0.129$ | 0.181 |
| $\tau$ | $-0.994$ | $-0.987$ | 0.104 | 0.148 |
| $N_{\text{int}}$ | $-0.471$ | $-0.431$ | 0.428 | $-0.290$ |
| $r_{\text{int}}$ | $-0.446$ | $-0.460$ | $-0.395$ | 0.344 |
| CBH | 0.148 | 0.063 | $-0.389$ | 0.752 |
| CTH | 0.143 | $-0.220$ | $-0.428$ | 0.765 |
| $H$ | $-0.795$ | $-0.812$ | $-0.200$ | 0.226 |
| $f_{\text{ad}}$ | $-0.284$ | $-0.273$ | 0.145 | 0.134 |

Table R11: Mean CRE ($\mathrm{W\,m^{-2}}$) for the LW radiation. Results are given as differences between the new scenario minus the reference simulation ($\Delta$). The root mean square error (RMSE) in $\mathrm{W\,m^{-2}}$ and the Pearson (Pears.) correlation between the new scenarios and the reference simulation are also given.

| Scen. | $\mathrm{CRE_{LW,B}}$ | | | $\mathrm{CRE_{LW,T}}$ | | |
|---|---|---|---|---|---|---|
| | $\Delta$ | RMSE | Pears. | $\Delta$ | RMSE | Pears. |
| S1a | $-0.11$ | 0.48 | 0.999 | $-0.04$ | 0.19 | 1.000 |
| S1b | $-0.05$ | 0.40 | 0.999 | $-0.03$ | 0.18 | 1.000 |
| S1c | $-0.01$ | 0.50 | 0.999 | $-0.01$ | 0.22 | 1.000 |
| S2a | 0.40 | 0.79 | 0.998 | 0.23 | 0.51 | 0.999 |
| S2b | 0.51 | 0.82 | 0.998 | 0.27 | 0.53 | 0.999 |
| S2c | 0.55 | 0.85 | 0.998 | 0.29 | 0.54 | 0.999 |
| S3a | $-0.05$ | 0.74 | 0.997 | 0.33 | 0.64 | 0.999 |
| S3b | $-0.01$ | 0.73 | 0.997 | 0.36 | 0.65 | 0.999 |
| S3c | 0.02 | 0.83 | 0.996 | 0.37 | 0.68 | 0.998 |
| S4 | $-0.02$ | 0.49 | 0.999 | $-0.02$ | 0.22 | 1.000 |